# The genomic architecture of circulating cytokine levels points to drug targets for immune-related diseases
Marek J. Konieczny [1], Murad Omarov [1], Lanyue Zhang[1], Rainer Malik[1], Tom G. Richardson [2,3], Sebastian-Edgar Baumeister[4], Jürgen Bernhagen [1,5,6], Martin Dichgans [1,5,6,7] & Marios K. Georgakis [1,5,8] ✉

Circulating cytokines orchestrate immune reactions and are promising drug targets for immune-mediated and inflammatory diseases. Exploring the genetic architecture of circulating cytokine levels could yield key insights into causal mediators of human disease. Here, we performed genome-wide association studies (GWAS) for 40 circulating cytokines in meta-analyses of 74,783 individuals. We detected 359 significant associations between cytokine levels and variants in 169 independent loci, including 150 *trans-* and 19 *cis*-acting loci. Integration with transcriptomic data point to key regulatory mechanisms, such as the buffering function of the *Atypical Chemokine Receptor 1 (ACKR1)* acting as scavenger for multiple chemokines and the role of *tumor necrosis factor receptor-associated factor 1 (TRAFD1)* in modulating the cytokine storm triggered by TNF signaling. Applying Mendelian randomization (MR), we detected a network of complex cytokine interconnections with TNF-b, VEGF, and IL-1ra exhibiting pleiotropic downstream effects on multiple cytokines. Drug target *cis*-MR using 2 independent proteomics datasets paired with colocalization revealed G-CSF/CSF-3 and CXCL9/MIG as potential causal mediators of asthma and Crohn's disease, respectively, but also a potentially protective role of TNF-b in multiple sclerosis. Our results provide an overview of the genetic architecture of circulating cytokines and could guide the development of targeted immunotherapies.

Chronic inflammation contributes to multiple human diseases, including allergic and autoimmune diseases, cardiometabolic diseases, and cancer. Inflammatory proteins like cytokines, chemokines and growth factors (hereafter collectively referred to as "cytokines") orchestrate the immune response underlying inflammation[1,2]. Circulating cytokines (i.e. cytokines measured in the circulation, including serum and blood specimens) are readily accessible and, therefore, attractive targets for therapeutic modulation, as they represent soluble ligands that execute downstream mechanisms through binding to membrane receptors or other circulating agents[3]. While several immunotherapies targeting circulating cytokines have been successfully introduced into the clinic, the lack of efficacy in other indications and the usually associated susceptibility to infection underscore the need for targeted approaches[4,5]. Prioritizing specific downstream mediators is critical

to minimize safety signals and ensure adherence to a life-long pharmacotherapy[5,6].

Recent advances in human genetics have enabled an in silico prioritization of drug targets[7,8], with approval rates more than two times higher than targets without genetic support[9]. Mendelian randomization (MR) uses data from genome-wide association studies (GWAS) and offers a statistical framework for exploring associations between variants in genes encoding drug targets and disease traits[10]. Previous MR analyses have illustrated the potential of integrating GWAS data for circulating proteins, including cytokines, with disease outcomes to discover novel drug targets[11–16]. However, existing efforts have been largely restricted by the small sample sizes of GWAS studies for circulating cytokines. For example, the largest-to-date targeted GWAS, which focused specifically on circulating cytokines

[1]Institute for Stroke and Dementia Research (ISD), LMU University Hospital, LMU Munich, Munich, Germany. [2]Medical Research Council (MRC) Integrative Epidemiology Unit (IEU), University of Bristol, Bristol, UK. [3]Population Health Sciences, Bristol Medical School, University of Bristol, Bristol, UK. [4]Institute of Health Services Research in Dentistry, University of Münster, Münster, Germany. [5]Munich Cluster for Systems Neurology (SyNergy), Munich, Germany. [6]German Centre for Cardiovascular Research (DZHKMunich), Munich, Germany. [7]German Center for Neurodegenerative Diseases (DZNE), Munich, Germany. [8]Program in Medical and Population Genetics, Broad Institute of MIT and Harvard, Cambridge, MA, USA. ✉e-mail: marios.georgakis@med.uni-muenchen.de

included up to 8293 individuals and allowed the detection of 27 significant genomic loci for 41 cytokines[17].

Novel proteomic platforms, such as the aptamer-based SOMAScan® and the proximity extension assay Olink®, gain popularity in quantifying at scale large numbers of proteins including cytokines. Here, we performed cross-assay comparisons in the genetic architecture of 40 cytokines quantified with three approaches (multiplex bead-based immunoassay, aptamer-based assay, proximity extension assay) and pooled data in GWAS meta-analyses including up to 74,783 individuals. This effort allowed the detection of 359 significant associations between 169 independent genomic loci and one or more of the 40 cytokines offering novel insights into mechanisms regulating circulating cytokine levels. Applying MR, we establish a causal cytokine network including upstream megaregulator cytokines that exert influence on a range of other cytokines. Finally, integrating these data with GWAS data for relevant disease endpoints, we provide genetic support for putative anti-inflammatory drug targets.

## Results

### Study cohorts and cross-assay reproducibility rate of significant genomic loci

We leveraged summary-level GWAS data for 40 circulating cytokines from three published datasets summing up to 74,783 individuals: the Cardiovascular Risk in Young Finns Study (YFS) and FINRISK studies that measured cytokines in serum using Luminex bead-based multiplex immunoassays (N = 8293); the Systematic and Combined AnaLysis of Olink Proteins (SCALLOP) study that measured cytokines in plasma using the proximity extension assay-based Olink® platform (N = 30,931); and the dataset provided by deCODE that measured cytokines in plasma using the aptamer-based SOMAScan® assay (N = 35,559, Fig. 1).

Given the known differences across the assaying methods, we first tested the replication rate of significant variants across all cytokines detected in each dataset in the other two datasets[18]. Although the GWAS in SCALLOP identified a considerably lower number of genome-wide significant loci for the available cytokines (n = 119 SNPs found for 13 cytokines), these variants exhibited the highest reproducibility rate (p < 0.05 and directionally

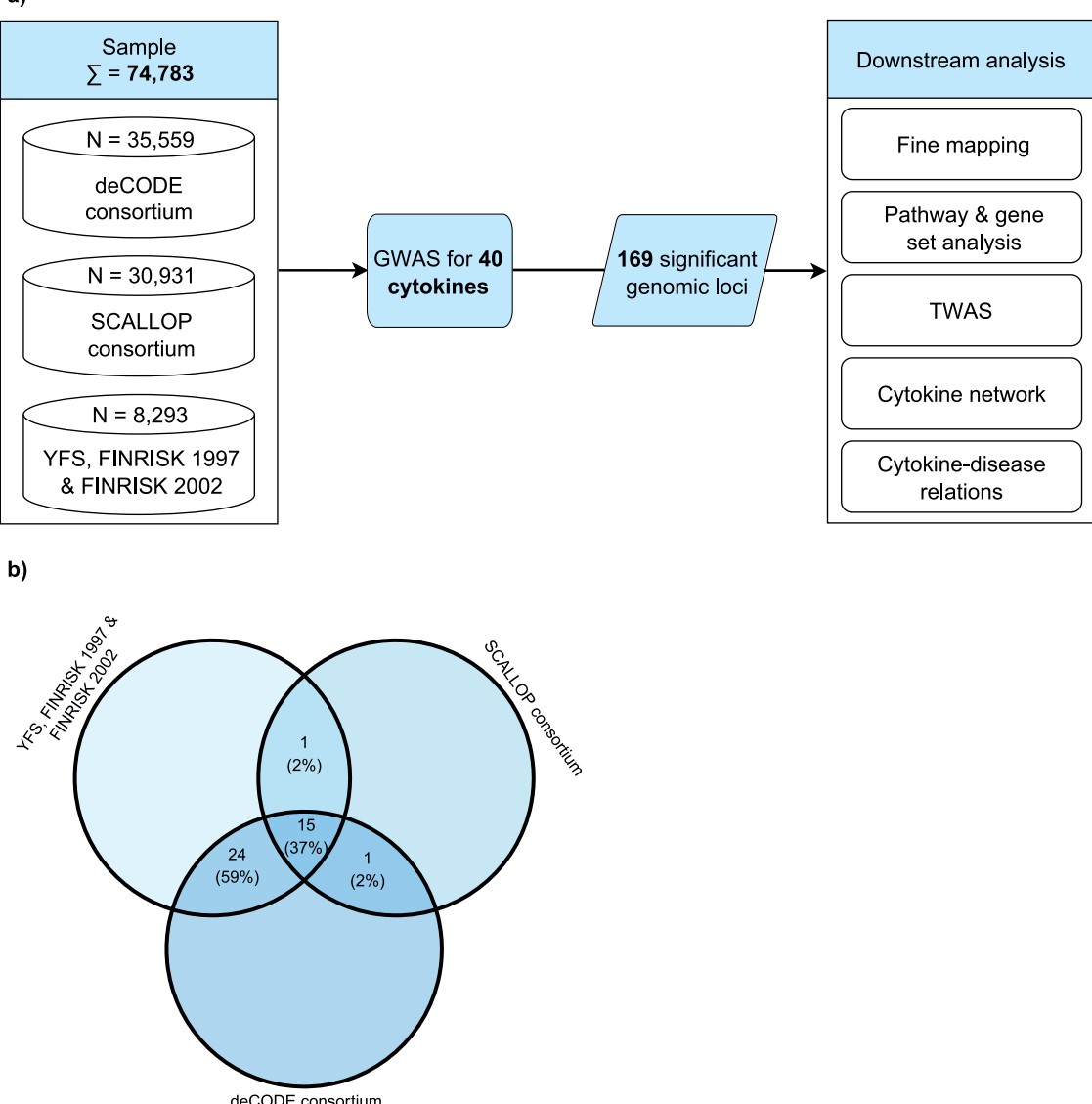

**Fig. 1 | Flowchart of the study design. a** Illustration of the analytical pipeline steps applied in this study to decipher the genetic architecture of circulating cytokines and their relation to allergic and autoimmune, cardiometabolic, and cancer outcomes. **b** Venn diagram shows the number (and percentage) of overlapping cytokines available in the three cohorts. SCALLOP Systematic and Combined AnaLysis of Olink Proteins, SNP single-nucleotide polymorphism, TWAS-MR transcriptome-wide Mendelian randomization analysis, YFS & FINRISK Cardiovascular Risk in Young Finns Study.

consistent) across the other two datasets. Specifically, 79 out of 119 variants replicated in YFS and FINRISK, with a median replication rate of 67%, and 46 out of 77 variants replicated in deCODE, with a median replication rate of 63% (Fig. 2, Supplementary Fig. S1, and Supplementary Data S1). In contrast, variants identified as significant in YFS & FINRISK had a lower replication rate in deCODE (70 out of 785 variants replicated, with a median replication rate of 4%) and in SCALLOP (101 out of 481 variants replicated, with a median replication rate of 11%). Similarly, variants identified in deCODE showed relatively low reproducibility with a median replication rate of 21% in YFS & FINRISK (141 out of 607 SNPs replicated) and 19% in SCALLOP (73 SNPs out of 252 replicated). The cytokines that showed the highest relative proportion of reproducible SNPs across all three datasets independently of the measuring assay were CC chemokine ligand 2 (CCL2/MCP-1) and vascular endothelial growth factor (VEGF).

## GWAS meta-analysis reveals novel *trans*- and *cis*-acting variants

Next, we performed GWAS meta-analyses across the three datasets. We identified a total of 359 significant associations between variants at 169 independent genomic loci and the circulating levels of one or more of the 40 cytokines ($p < 5 \times 10^{-8}$ in fixed-effects meta-analysis, Fig. 3, Supplementary Data S2). Variants that showed significant heterogeneity between the three cohorts (HetPval < 0.1) are reported in Supplementary Data S2 (48% of the significant loci with HetPval < 0.1; range 0–100% across 40 cytokines and 41% of the significant loci with HetPval < 0.05; range 0–100% across 31 cytokines). The lambda values ranged between 0.96 for interleukin (IL)-16 and 1.04 for basic fibroblast growth factor (FGF-b), indicating the absence of overall inflation in the test statistics (Supplementary Data S2). According to the GWAS catalog (https://www.ebi.ac.uk/gwas/)[19], 156 of the loci have not been associated with circulating levels of the 40 cytokines in previous GWASs (Supplementary Data S2). Assessing the loss of loci that were significant in the original studies, we found that 18% of loci lost significance in the meta-analyzed samples of the three source databases[17,20,21]. The proportion of explained variance by the significant variants ranged from 0.0008 for IL-17 to 0.033 for stem cell growth factor beta (SCGF-b) (Supplementary Data S2).

As expected, due to larger effect sizes for rare genetic variants, we found a strong inverse correlation between minor allele frequency and effect size (Spearman's rho = −0.827, $p = 5 \times 10^{-30}$, Fig. 4a). The majority of the significant loci (150 out of 169) represented *trans*- (distant-) acting variants. When excluding the human leukocyte antigen (HLA) region on chromosome 6, we found 33 pleiotropic variants showing associations with >1 cytokine among the significant *trans*-variants (Fig. 4b). A locus hotspot associated with multiple cytokines was found at the region of the gene encoding *complement factor H* (*CFH*). This soluble mediator plays an essential role by interacting with the C3 convertase for the regulation of inflammatory responses exerted by the complement system, which could possibly explain the associations with multiple cytokines[22]. While at least 1 significant *trans*-variant was present for all studied cytokines (median number of variants per cytokine = 5, range 1–2), we found significant *cis*- (local-) acting variants in the vicinity of their encoding gene for 19 cytokines (Supplementary Data S2). The lead *cis*-acting variants showed stronger associations with cytokine levels (mean absolute beta: 0.18, range: 0.05–0.94) than *trans*-acting variants (mean absolute beta: 0.08, range: 0.03–0.55, p-for-comparison = 0.03, Fig. 4c).

To map causal variants responsible for associations between circulating cytokine levels and genes within each of the 169 independent genomic loci we used SuSiE fine mapping. Employing a Bayesian framework, fine mapping identifies credible sets of variants with a posterior probability (PP) of 95%. The number of variants within credible sets ranged from 2 to 50. The highest numbers of variants within a credible set were found at 15q21.3 for stem cell factor (SCF) (n = 50), at 19q13.33 for SCGF-b (n = 49), and at 6p21.1 for VEGF (n = 44). SuSiE mapped the association test lead variant to the credible sets for 49 genomic loci, identifying the GWAS lead as the most likely causal mutation (Supplementary Data S3).

## Functional follow-up analyses highlight immune response regulatory mechanisms

To understand the biological significance and downstream functional impact of the identified variants, we performed follow-up analyses. We performed a gene-based multi-marker analysis of genomic annotation (MAGMA) analysis, which combines the effects of multiple SNPs to identify associations between genes and circulating cytokine levels. This analysis showed 829 significant associations with the levels of circulating cytokines at a Bonferroni-defined significance level (Supplementary Data S4). In total, 626 uniquely mapped genes were associated with at least 1 cytokine. The number of genes mapped to cytokines ranged from 1 for beta nerve growth factor (bNGF), cutaneous T-cell attracting (CCL27/CTACK), IL-10, and tumor necrosis factor-alpha (TNF-a) to up to 95 genes mapped for SCGF-b, 92 genes for macrophage inflammatory protein-1β (CCL4/MIP-1b) and 51 genes for CCL11/eotaxin-1. In line with our GWAS results, the gene that was mapped for most cytokines (n = 16) was *CFH*. The genes with the lowest *p*-value were *H4 Clustered Histone 14 (H4C14)* for monocyte-specific chemokine 3 (CCL7/MCP-3) ($p = 1 \times 10^{-50}$), *Ribosomal Protein S17 (RPS17)* for IL-16 ($p = 1.3 \times 10^{-45}$), and *ATP Binding Cassette Subfamily A Member 1 (ABC1)* for SCF ($p = 1.9 \times 10^{-29}$). A gene-property analysis revealed that the cytokine-related genes were primarily enriched for expression in the liver ($p = 4.9 \times 10^{-10}$), in line with its well-established role as a main source of production of many cytokines. Other enriched tissues included the spleen ($p = 4.9 \times 10^{-4}$) and lung ($p = 5.9 \times 10^{-4}$, Supplementary Data S4). Further combining the genes to sets related to concrete biological pathways, we performed a gene-set MAGMA analysis which prioritized 41 pathways for 12 cytokines that reached a Bonferroni-adjusted significance level ($p < 1.2 \times 10^{-7}$, Supplementary Data S4). The identified pathways were primarily related to immune response with a small cluster involved in metabolic and developmental processes.

Positional mapping ascribed 75% of significant variants to intronic (54%) and intergenic (21%) regions, suggesting that the identified variants primarily determine gene transcription or gene expression profiles (Supplementary Data S2)[23]. Thus, we integrated our GWAS data with transcriptomic data and performed a transcriptome-wide association study (TWAS) using Mendelian randomization (MR) for a deeper elaboration on the transcriptional effects underlying our GWAS results. Summary statistics for expression quantitive trait loci (eQTLs) in whole blood were obtained from the eQTLGen consortium including transcriptomic profiles for 31,684 individuals of primarily European ancestry[24]. Using *cis*-eQTLs as genetic instruments, we identified 245 significant associations between genetically proxied gene expression in whole blood and cytokines levels. Sensitivity analyses showed directional concordance for 77% and 76% of the associations calculated with weighted median MR and MR Egger, respectively, and for weighted median MR 62% of the associations were significant (Fig. 5a, Supplementary Fig. S5 and Data S5). The number of significant genes per cytokine ranged from 1 to 18. While most significant genes (78%) influenced the levels of a single cytokine, the genetically proxied expression of 54 genes showed an effect on circulating levels of up to 9 cytokines (n[*SKI2 Subunit Of Superkiller Complex; SKIV2L*] = 9, n[*Major Histocompatibility Complex, Class II, DR Beta 5; HLA-DRB5*] = 9, n[*Negative Elongation Factor Complex Member E; NELFE*] = 7, n[*Atypical Chemokine Receptor 1; ACKR1*] = 5, n[*Fc Epsilon Receptor Ia; FCER1A*] = 4, n[*TNF receptor-associated factor-Type Zinc Finger Domain Containing 1; TRAFD1*] = 4, n[*Leucine Carboxyl Methyltransferase 2; LCMT2*] = 4) (Fig. 5b). Interestingly, we found significant *cis*-effects of the encoding gene expressions on the circulating levels of only 3 of the 40 respective cytokines. This is in line with previous eQTL–pQTL comparisons and aligns with the fact that the circulating proteome is not the direct product of the whole-blood transcriptome[25,26].

Excluding genes within the very dense HLA region (i.e. *SKIV2L*, *HLA-DRB5* and *NELFE*) we explored deeper the biological relevance of the pleiotropic *ACKR1*, *TRAFD1* and *LCMT2* genes. The genetically proxied mRNA levels of *ACKR1* were associated with circulating CCL2/

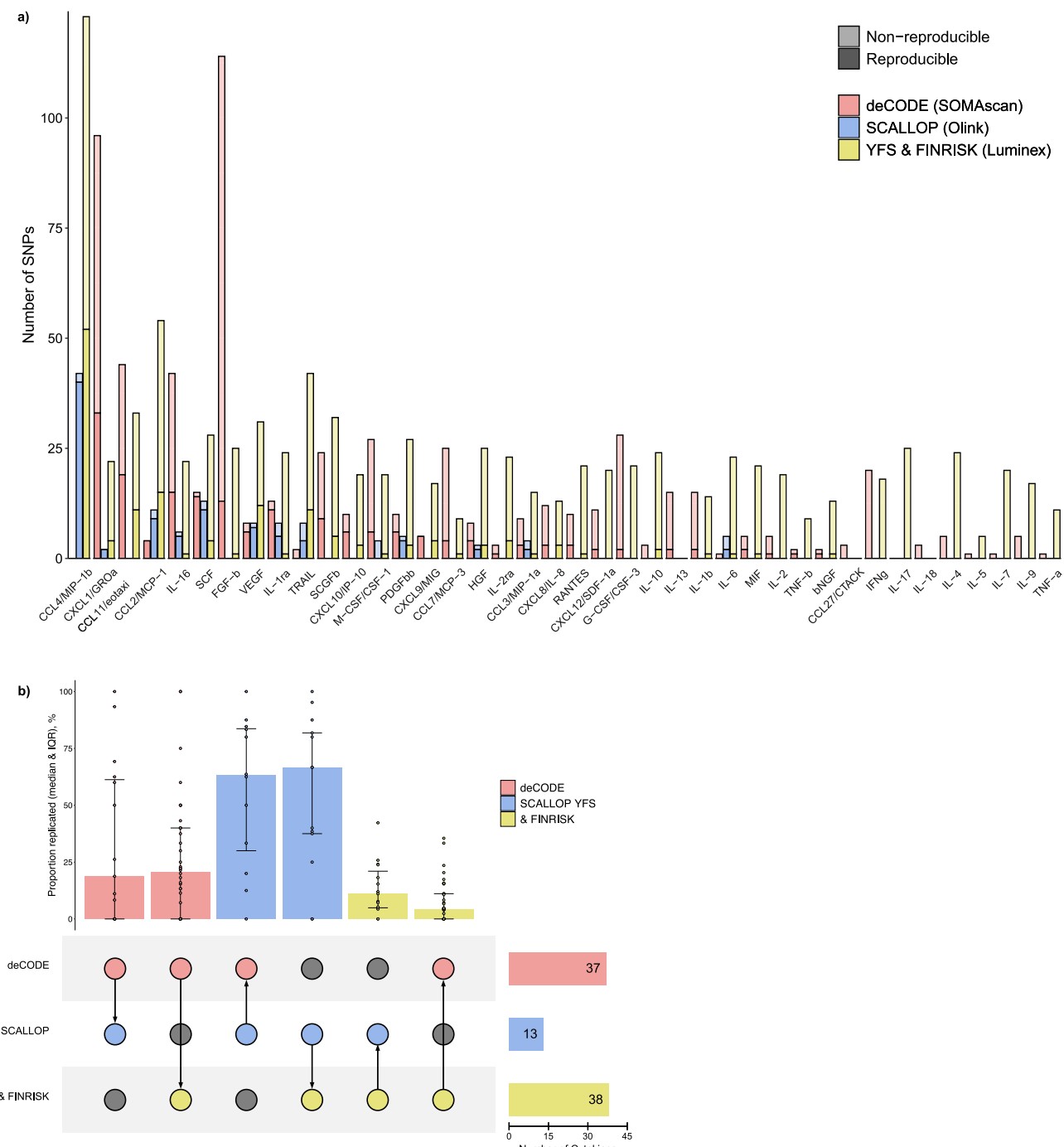

**Fig. 2 | Comparisons of significant genomic loci for 40 circulating cytokines across three proteomics assays. a** Number of reproducible and non-reproducible SNPs per cytokine (depicted as saturated and light-colored bars, respectively) for deCODE, SCALLOP, and YFS & FINRISK cohorts. The saturated portion of the bars represents the number of SNPs that were replicated in at least one other cohort, where reproducibility is defined as SNPs confined to significant loci (*p*-value < 0.05) and directionally concordant. **b** Proportion of replicated SNPs across cytokine datasets visualized for all possible combinations of cohorts. The matrix at the bottom left shows the comparison each of the vertical bars at the top represents. Arrows in the comparison matrix illustrate the direction of comparison—from the reference dataset where significant SNPs were identified, to the dataset in which SNPs were replicated. For example, the first bar displays the percentage of loci in deCODE that were replicated in SCALLOP (with the arrow pointing from deCODE to SCALLOP). The horizontal bars on the bottom right show the number of cytokines for which significant SNPs were found. The sample sizes used to derive the statistics in **b** are from left to right 73, 141, 46, 79, 101, and 70. Median, IQR (error bars represent the 25th and 75th percentiles). Colored bars represent the deCODE consortium in red, the SCALLOP consortium in blue and YFS & FINRISK cohorts in yellow. IQR interquartile range, SNP single-nucleotide polymorphism, SCALLOP Systematic and Combined AnaLysis of Olink Proteins, YFS & FINRISK Cardiovascular Risk in Young Finns Study.

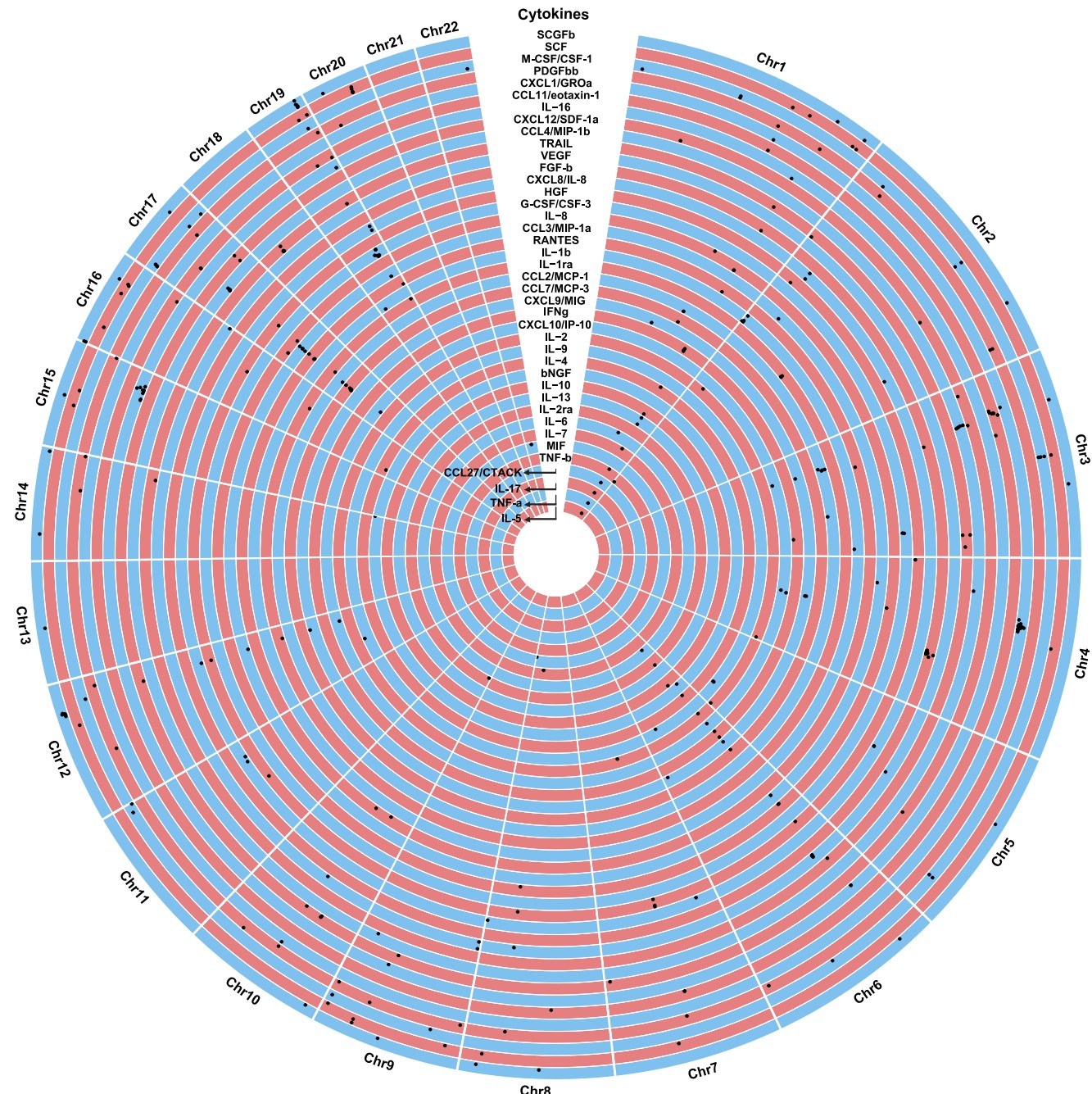

**Fig. 3 | Genetic architecture of the circulating levels of the 40 cytokines.** Circular Manhattan plot of genomic loci significantly associated with circulating levels of 40 cytokines in a meta-analysis of the three datasets. The 359 genome-wide significantly associated variants at $p < 5 \times 10^{-8}$ are depicted as black dots for GWAS meta-analyses in YFS & FINRISK, SCALLOP, and deCODE cohorts. The horizontal and vertical locations of dots in each single rectangle depict genomic positioning (increasing from left to right) and $p$-value (decreasing from bottom to top), respectively. SCALLOP Systematic and Combined AnaLysis of Olink Proteins, YFS & FINRISK Cardiovascular Risk in Young Finns Study.

MCP-1, CCL7/MCP-3, CCL11/eotaxin-1, growth regulated oncogene-α (CXCL1/GROa), CXCL8/IL-8 (Fig. 5b). *ACKR1* codes for a cell-surface receptor that binds, internalizes and transports multiple CC and CXC chemokines and promotes leukocyte transcytosis into the circulation[27,28]. By acting as a scavenger receptor, ACKR1 modulates the bioavailability of cytokines and thereby affects inflammatory responses[29,30]. The identified associations were driven by rs12075, a well-characterized missense variant in *ACKR1*, resulting in less efficient chemokine binding to ACKR1 due to the loss of a necessary amino-acid sulfation (Fig. 6a)[31]. The impaired receptor binding leads to elevated circulatory levels of chemokines and might, in turn, result in a compensatory increase in

*ACKR1* expression, which could explain the positive association between genetically proxied *ACKR1* and its ligands[32]. We replicated previously reported associations between *ACKR1* and levels of CCL2/MCP-1, CCL7/MCP-3, CCL11/eotaxin-1, and CXCL1/GROa and additionally showed an association with CXCL8/IL-8 levels[17,25]. The genetically proxied expression of *TRAFD1* was also associated with multiple circulatory cytokine levels, including CCL7/MCP-3, monokine induced by interferon-gamma (CXCL9/MIG), interferon gamma-induced protein 10 (CXCL10/IP-10), and tumor necrosis factor-beta (TNF-b) (Fig. 5b). *TRAFD1* functions as an adaptor protein that binds to the intracellular domain of TNF receptors expressed on both innate and adaptive

**Fig. 4 | *Trans*- and *cis*-acting genetic variants underlying circulating cytokines. a** Inverse correlation between minor allele frequency and effect size is illustrated for *trans*- and *cis*-acting loci, depicted as gray and red circles, respectively. **b** Number of significant loci binned by the number of associated circulating cytokines (excluding the HLA region on chromosome 6). **c** *Cis*-acting variants (*n* = 19, red bar) showed stronger associations with cytokine levels when compared with *trans*-acting variants (*n* = 150, gray bar). Bars and lines represent median and 95% confidence intervals, respectively. HLA human leukocyte antigen, SD standard deviation.

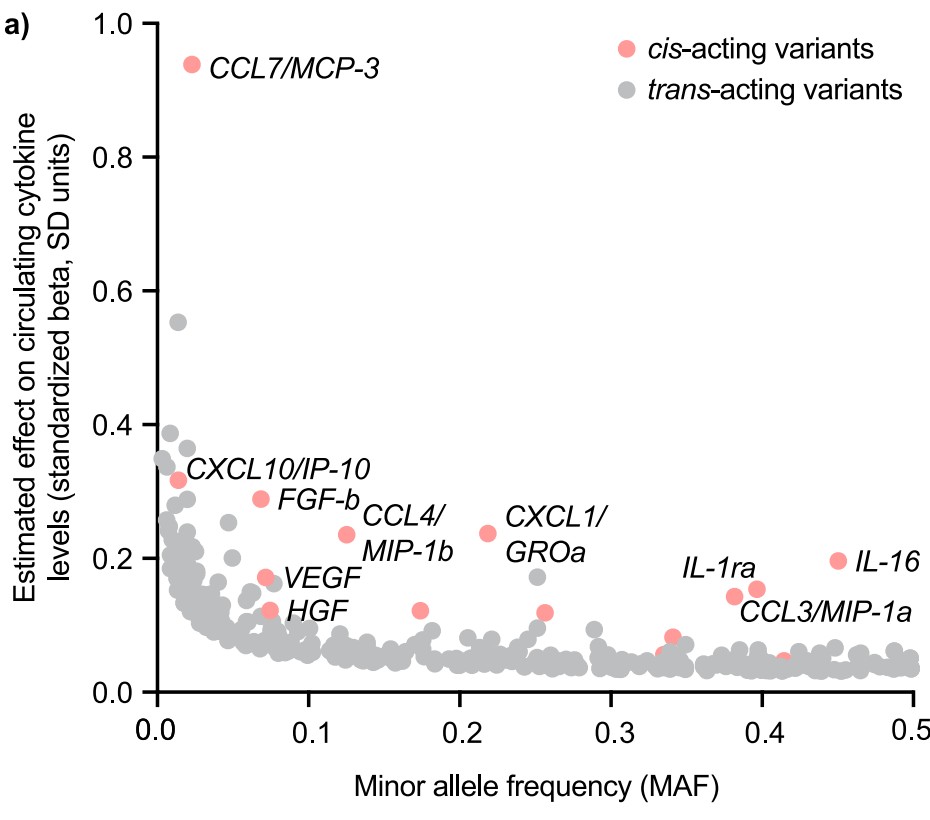

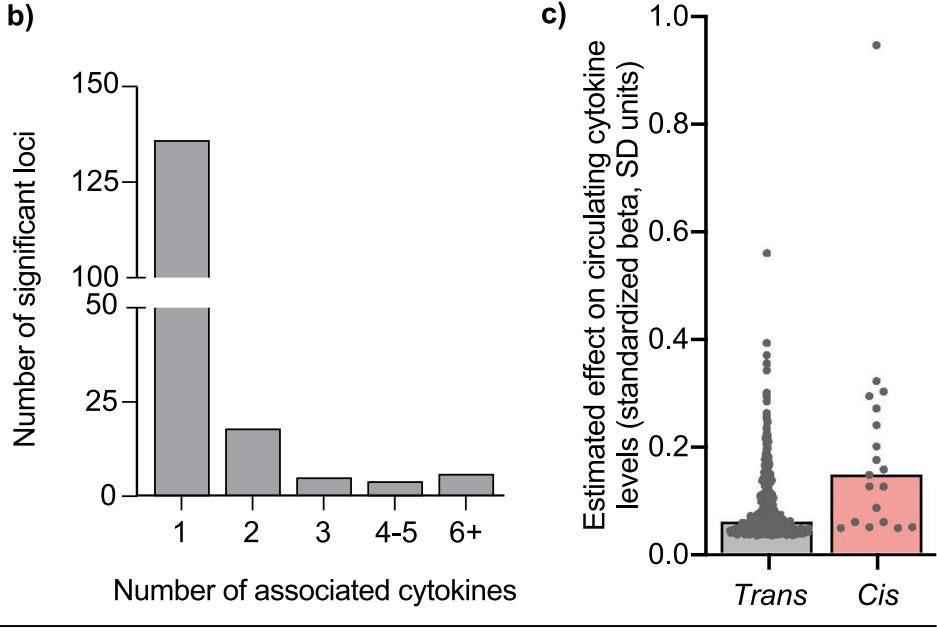

immune cells. It regulates downstream signaling also involving the NF-κB pathway and thereby modulates the production of several pro-inflammatory cytokines and inflammatory responses[33–35] (Fig. 6b). *TRAFD1* is a master regulator of genes involved in interferon-γ (IFNg) signaling and T-cell receptor activation[36]. Genetically proxied expression of the gene encoding for *LCMT2* showed associations with beta nerve growth factor (bNGF), CXCL8/IL-8, CXCL10/IP-10, and platelet-derived growth factor-bb (PDGFbb). *LCMT2* is involved in amino-acid metabolism, presumably regulating hypothalamic gene expression but there is only limited knowledge on its biological function[37,38].

**Genetic associations point to network interactions between circulating cytokines**

As a next step, we explored cross-trait genetic correlations between the circulating levels of the 40 studied cytokines (Fig. 7a, Supplementary Data S6). One-third of the between-cytokine correlations were significant at *p* < 0.05; the vast majority of the significant associations (96%) were positive. Due to a computation error in 5% of cross-cytokine interactions (80 of 1600) caused by missing evidence of SNP-heritability for two phenotypes (MIF [Macrophage migration inhibitory factor] and G-CSF/CSF-3) LDSC values and therefore correlations could not be computed. Furthermore, to

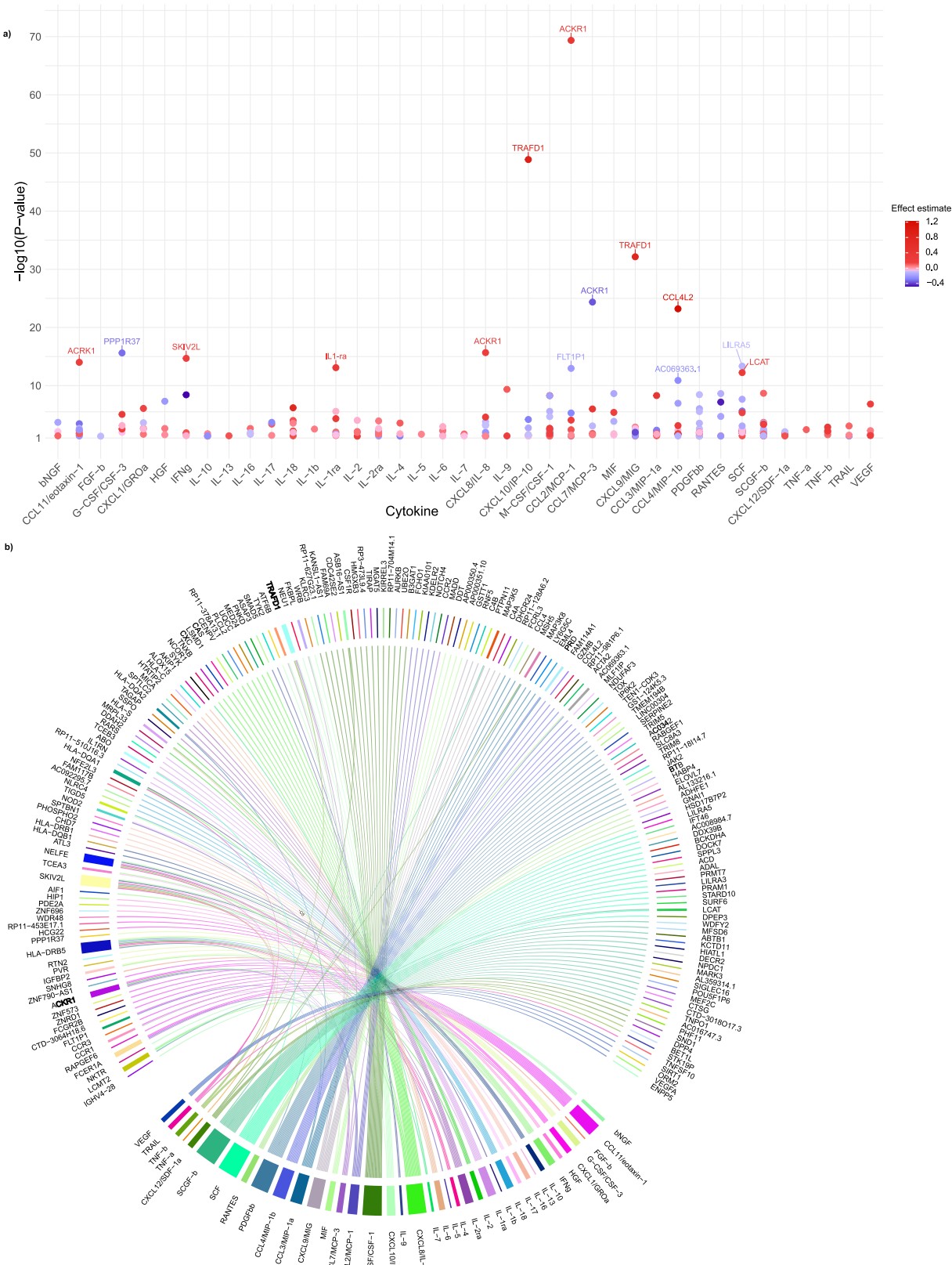

**Fig. 5 | Genetically predicted gene expression in peripheral blood partly explains the genetic architecture of 40 circulating cytokine levels. a** The dots represent genes, the blood expression of which was significantly associated with circulating cytokine levels in a Mendelian randomization-based transcriptome-wide association study. Short names of genes are depicted for the top-line results ($\log_{10} P$-value > 10). **b** Chord diagram visualizes the pleiotropic effects of genes (in the upper part of the figure) affecting circulating levels of cytokines (in the lower part of the figure).

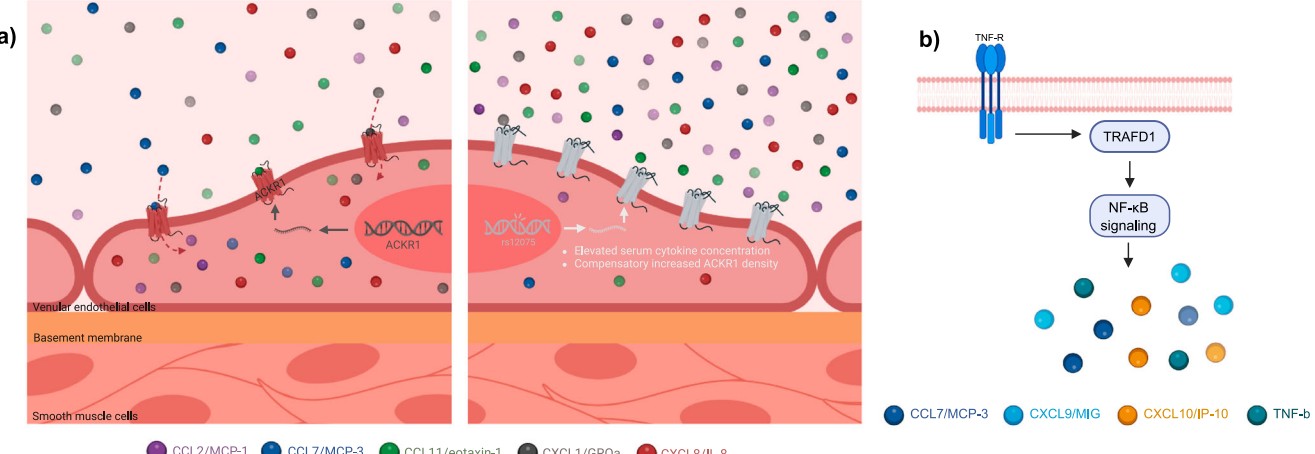

**Fig. 6 | Gene expression of ACKR1 and TRAFD1 exert pleiotropic effects on multiple cytokine levels. a** Schematic illustrating the impact of cis-eQTLs for ACKR1 on receptor function. The left-hand side shows the ACKR1 gene encoding the atypical chemokine receptor 1 functioning as a sink for multiple chemokines, which are buffered intracellularly in venular endothelial cells. Depicted on the right, is the missense variant rs12075 coding for a dysfunctional receptor with less efficient chemokine binding efficacy. This leads to higher levels of circulating CCL2/MCP-1, CCL7/MCP-3, CCL11/eotaxin-1, CXCL1/GROa, and CXCL8/IL-8 and possibly to a compensatory increase in ACKR1 expression and receptor density. Created in BioRender. User1, L. (2025) https://BioRender.com/p20h611. **b** Schematic illustrating how genetically proxied TRAFD1 expression regulates multiple cytokine levels (CCL7/MCP-3, CXCL9/MIG, CXCL10/IP-10, TNF-b), supporting its regulatory role in TNF-mediated NF-κB signaling. ACKR1 atypical chemokine receptor 1, TNF-R TNF-receptor, TRAFD1 tumor necrosis factor receptor-associated factor-type zinc finger domain-containing protein 1. Created in BioRender. User1, L. (2025) https://BioRender.com/k48o152.

understand causal interconnections between circulating cytokines levels, we performed MR analysis using *cis*-variants from our GWAS meta-analysis. We found significant (FDR-corrected $p < 0.05$) associations between 65 cytokine pairs (53 positive associations and 12 negative associations, Fig. 7b, sensitivity analyses excluding variants showing significant heterogeneity between the three datasets in Supplementary Fig. S2 and Supplementary Data S6). In pleiotropy-robust sensitivity analyses (weighted median, MR Egger) 89% and 85% of the 65 cytokine pairs showed directionally concordant associations, respectively (Supplementary Data S6). Genetically proxied levels of CCL7/MCP-3, stromal cell-derived factor-1alpha (CXCL12/SDF-1a), granulocyte colony-stimulating factor (G-CSF/CSF-3), IL-9, TNF-b, and VEGF were positively associated with the levels of >2 other cytokines, whereas genetically proxied CXCL1/GROa and IL-1 receptor antagonist (IL-1ra) were negatively associated with lower level of >2 other cytokines. Most significant associations were detected for TNF-b ($n = 13$), VEGF ($n = 9$), IL-1ra ($n = 7$), IL-9 ($n = 7$), and G-CSF/CSF-3 ($n = 7$). The negative associations between IL-1ra and several proinflammatory cytokines (CCL7/MCP-3, IL-9, TNF-a, TNF-b), chemokines (macrophage inflammatory protein-1α, CCL3/MIP-1a), and growth factors (hepatocyte growth factor, HGF; VEGF) align well with the immunoregulatory role of the IL-1 pathway and the inhibitory effect of IL-1ra on downstream IL-1 signaling[39,40]. TNF-b emerged as a significant player in our network analysis, demonstrating characteristics of a master regulator by showing significant associations with higher circulating levels of 13 mostly pro-inflammatory cytokines. Furthermore, TNF-b exhibited significant positive LDSC genetic correlations with 7 of the 13 cytokines, suggesting a shared genetic architecture within the TNF-β network (Fig. 7a). For both master regulator cytokines (i.e. TNF-b and IL-1ra) sensitivity analyses confirmed directional consistency for all associations using the two MR methods reported above (Supplementary Data S6). While certain interactions with TNF-b, such as those involving IL-1ra, TNF-a, TNF-related apoptosis-inducing ligand (TRAIL), and VEGF, are well-documented, the majority of interactions have not been reported previously and merit additional investigation[41–43].

### *Cis*-Mendelian randomization and colocalization highlight potential drug targets for immune-related diseases

For insights into the clinical consequences of genetically proxied levels of the circulating cytokines, we analyzed associations with allergic and autoimmune, cardiometabolic, and cancer outcomes in two-sample MR followed by colocalization analyses (Fig. 8a, sensitivity analyses excluding variants showing significant heterogeneity between the three datasets in Supplementary Fig. S4 and Supplementary Data S7). We used *cis*-acting genetic variants as instruments due to their lower likelihood of influencing cytokine levels through pleiotropic mechanisms. We further complemented these analyses with Bayesian colocalization to prioritize associations less likely to be influenced by pleiotropy due to linkage disequilibrium of studied variants with neighboring genes[44]. Following correction for multiple comparisons, we found 24 significant MR associations between genetically proxied cytokine levels and disease outcomes (14 positive and 10 negative associations). Sensitivity analyses showed directional concordance for 91% and 75% of the associations calculated with weighted median MR and MR Egger, respectively (Supplementary Fig. S6 and Data S7). Our MR findings partially confirmed established pathogenetic associations with diseases and therapeutic drug targets that are already in clinical application. For example, there is solid evidence linking IL-2 receptor subunit alpha (IL-2ra) increasing variants to elevated risk for multiple sclerosis (MS) and Crohn's disease (CD)[17] Aldesleukin, a recombinant form of IL-2 approved for cancer indications, is currently under investigation in a phase-2 clinical trial for CD (ClinicalTrials.gov ID: NCT04263831)[45]. Also, compounds targeting IL-1 signaling, anakinra or canakinumab, represent established treatment algorithms for inflammatory joint diseases like rheumatoid arthritis (RA) or juvenile arthritis[46,47].

Of the 24 signals, 4 also showed evidence of significant colocalization, that is a PP of association >80% for shared causal variants between cytokine levels and disease outcomes (Fig. 8a, Supplementary Data S7), thus providing even stronger evidence for causality. These included associations of higher genetically proxied G-CSF/CSF-3 levels with asthma, lower genetically proxied G-CSF/CSF-3 and higher genetically proxied CXCL9/MIG levels with CD, as well as lower genetically proxied TNF-b levels with MS.

Furthermore, the association between genetically proxied IL-1 receptor antagonist (IL-1ra) levels and lower risk of RA reached a PPA of 68% for a shared causal variant in colocalization analysis. These results are consistent with data from preclinical studies[48–50], observational studies in humans[51–55], and clinical trials[56,57], thus providing support for potentially promising targeted immunotherapies for these indications.

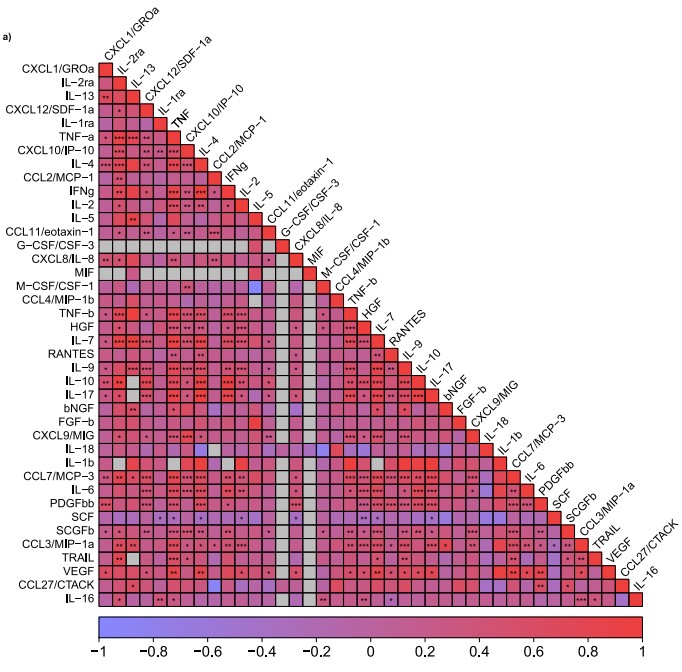

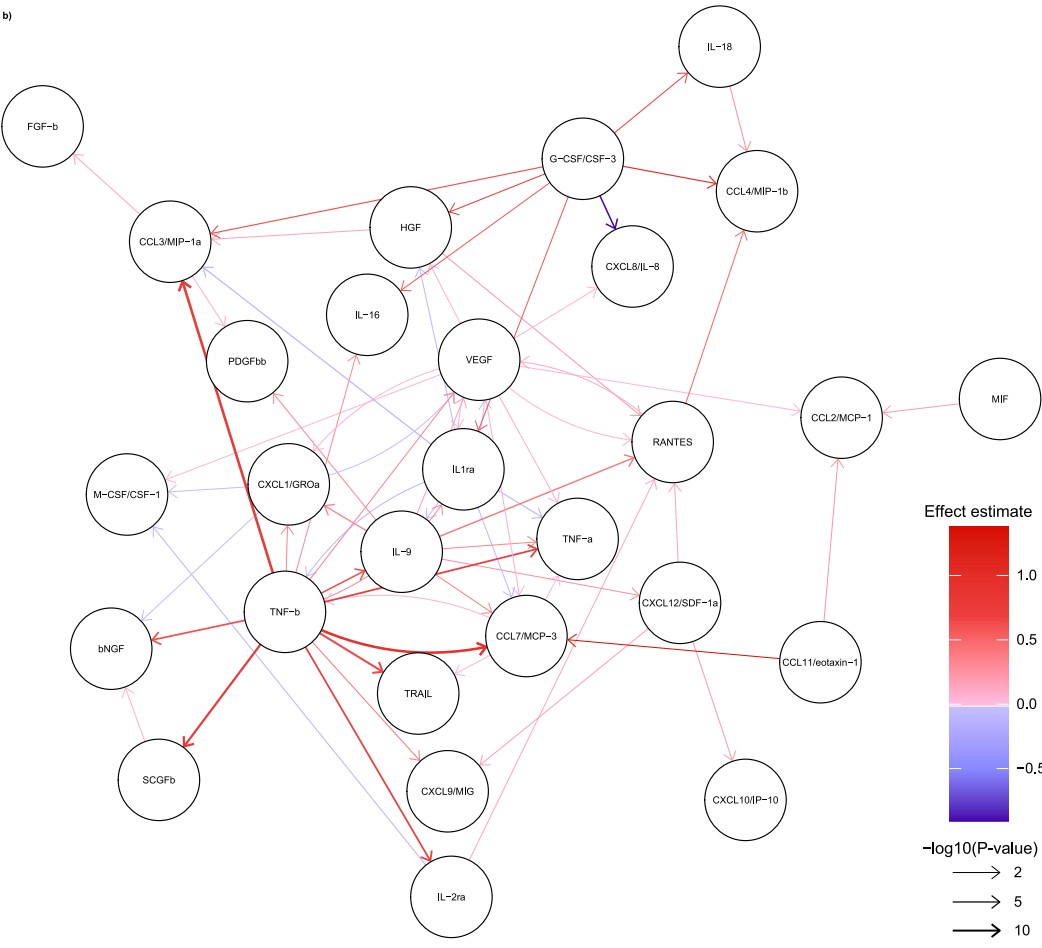

**Fig. 7 | Cross-cytokine genetic associations. a** Genetic correlations ($n = 1600$) with LD-score regression across cytokine serum levels are depicted as a correlation heatmap. Stars highlight significance level *0.05; **0.0001; ***0.00001. LD-score correlation coefficients are illustrated according to the legend below spanning from −1 in blue to +1 in red, missing correlation coefficients are depicted in gray and are due to no evidence of SNP-heritability for one of the cytokines. **b** *Cis*-Mendelian randomization excluding variants associated with the exposure and outcome in the instruments lists between genetically proxied circulating cytokine levels ($n = 65$). Arrow heads show the direction of causal influence, color gradient indicates the effect estimate, and line width is the logarithm-adjusted Benjamin–Hochberg corrected significance level. LD linkage disequilibrium.

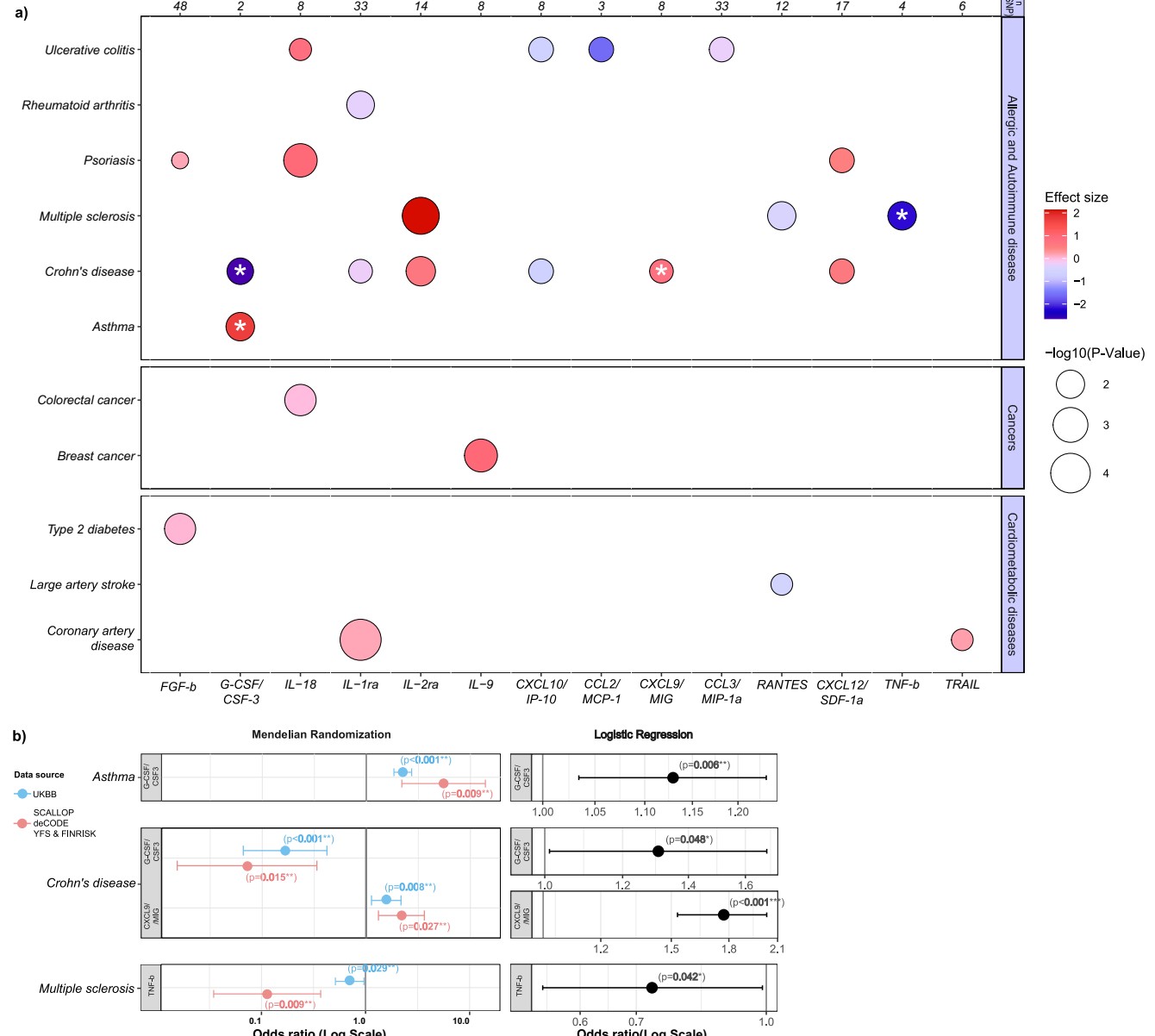

**Fig. 8 | Cis-Mendelian randomization associations and colocalization analyses between genetically proxied cytokine levels and disease risk. a** Significant associations between circulating cytokine levels and disease outcomes are shown for allergic and autoimmune, cardiometabolic, and cancer outcomes. Effect sizes and log-transformed, Benjamin–Hochberg corrected *p*-values are illustrated by color gradient and circle size, respectively. Only cytokines and disease endpoints with at least 1 significant association are depicted. Stars highlight significant genetic colocalizations (posterior probability of association > 80%) for shared causal variants between circulating cytokine levels and disease risk. n(SNP), indicates number of *cis*-acting genetic variants used as instruments in MR analyses. **b** Inlet from (**a**) for colocalized associations between G-CSF/CSF-3 and Asthma, G-CSF/CSF-3 and CD, CXCL9/MIG and CD, and TNF-b and MS. *Cis*-MR undertaken in the current meta-analysis (red colored) and UKBB proteomics cohort (light-blue colored) as well as logistic regression prediction model using longitudinal UKBB clinical data showing converging evidence. Error bars represent 95% confidence intervals. Log scale logarithmic scale, OR odds ratio, UKBB United Kingdom Biobank.

To substantiate our MR results we validated our findings with genetic instruments derived from the independent United Kingdom Biobank (UKBB) data. Using *cis*-acting variants for G-CSF/CSF-3, CXCL9/MIG, and TNF-b from the UKBB proteomics dataset as instruments, we could confirm significant associations with asthma, CD, and MS, respectively. Furthermore, we performed logistic regression analyses for these cytokines and the respective disease endpoints defined by ICD-9 and ICD-10 codes in the UKBB with three of the associations being significant and directionally consistent with the MR results (Fig. 8b). The c-index of an age- and sex-adjusted model for asthma and multiple sclerosis did not significantly improve after adding G-CSF (0.558–0.562) and TNF-b levels (0.612–0.618), respectively, but the

addition of MIG levels led to significant prediction gains for Crohn's disease (0.536–0.659).

## Integration of cytokine-disease MR and TWAS-MR results implicates additional mediators of disease mechanisms that could represent promising drug targets

As a last step, we aimed to integrate the cytokine-disease MR results with the TWAS MR results with the goal of also detecting upstream regulators of the potentially causal cytokines. We performed MR analyses between genetically proxied expression of genes significantly associated with G-CSF/CSF-3, CXCL9/MIG, and TNF-b in our TWAS-MR analyses and the associated disease outcomes. We found that higher genetically

**Fig. 9 | Causal associations between genetic regulators for cytokines, circulating cytokine levels, and disease risk. a** Genetically proxied mRNA for *PPP1R37*, *PVR*, *RTN2*, and *IGFBP2* affect circulating G-CSF/CSF-3 levels leading to increased risk for asthma. In turn, *PPP1R37* directly lowers disease risk for asthma. Created in BioRender. User1, L. (2025) https://BioRender.com/r76a901.
**b** Genetically proxied mRNA for 11 genes underlying CXCL9/MIG levels differentially affect circulating cytokine concentrations which influence the risk for Crohn's disease. Independently, *TRAFD1*, *ATF6B*, and *C4A* also modulate disease risk for Crohn's disease. Created in BioRender. User1, L. (2025) https://BioRender.com/r76a901.

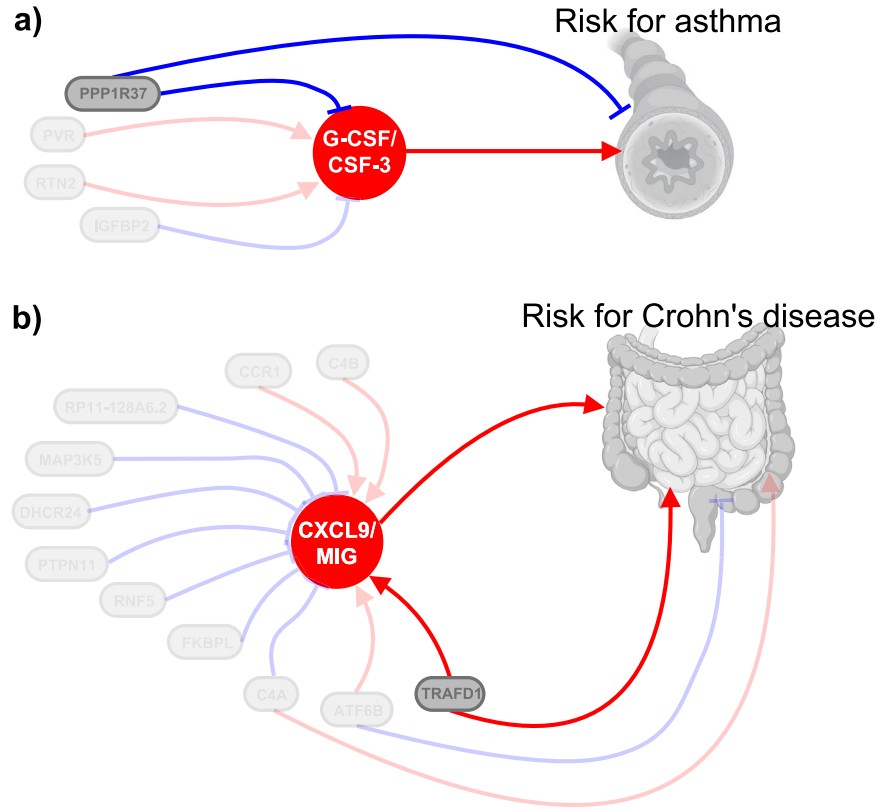

proxied expression of *PPP1R37* is associated with lower levels of G-CSF/CSF-3, as well as with a lower risk of asthma (Fig. 9a). We also found higher genetically proxied expression of *TRAFD1* to be associated with higher CXCL9/MIG levels and higher risk of CD (Fig. 9b).

## Discussion

Pooling data from up to 74,783 patients from three independent GWAS cohorts, we identified 169 independent genomic loci influencing the circulating concentration of 40 cytokines, 156 of which have not been associated with circulating cytokine levels in previous GWASs. Integrating our results with transcriptomic data, a TWAS-MR analysis revealed 245 potentially causal associations between gene expression of mostly immunoregulatory genes in peripheral blood and circulating cytokine levels. Analyzing regulatory interactions between cytokines, we found TNF-b, VEGF, and IL-1ra as master controllers of the circulating levels of multiple cytokines. Finally, we provide genetic evidence based upon two independent proteomics datasets that the circulating levels of three cytokines (G-CSF/CSF-3, CXCL9/MIG, TNF-b) might be causally involved in the pathogenesis of asthma, CD, and MS, thus offering insights for the development of more specific immunotherapies. We backed the genetic data with clinical data from UKBB showing predictive biomarker properties for the four cytokine–disease pairings.

Our MR and colocalization analyses provided evidence for potential causality for four cytokine–indication pairs, thus offering genetic support for potentially promising targeted immunotherapies for asthma, CD, and MS. These results are highly consistent with preclinical, epidemiological, and occasionally clinical data. For example, G-CSF/CSF-3 is a pro-inflammatory cytokine involved in neutrophil differentiation and systemic mobilization and has been implicated in the pathogenesis of neutrophilic atopic asthma[48,58]. Several preclinical studies in asthma models showed that blockage of upstream inductors or the receptor of G-CSF/CSF-3 reduced circulating cytokine levels, alleviated the airway inflammatory response, and improved disease outcome[48–50,59]. Furthermore, G-CSF/CSF-3 levels in the sputum of asthma patients have been suggested as a marker of airway neutrophilic inflammation[60]. Our results provide genetic support for the concept of targeting G-CSF/CSF-3 in asthma, potentially focusing on patients with neutrophilic asthma.

Using integrated results from our GWAS and TWAS findings, we identified an upstream mechanism of yet unknown relevance: higher *PPP1R37* gene expression was associated with reduced G-CSF/CSF-3 levels and lower risk of asthma. The gene encodes a regulatory subunit that acts as a phosphatase inhibitor and has, so far, not been associated with airway diseases[61]. Previous studies investigating related regulatory subunits have unveiled potential biological mechanisms through which these subunits may influence the immune response[62,63], and genetic studies have provided support for the significance of other protein phosphatase regulatory subunits as contributing factors to airway diseases[64–66]. For example, genetically proxied expression of *PPP1R3D* was associated with disease characteristics of asthma, including mucosal immunity, cell metabolism, and airway remodeling, and predicted responsiveness to omalizumab therapy[64]. Although the specific biological mechanism underlying our finding is unknown, one might speculate that the diverse range of functions associated with PP1, including cell progression, apoptosis, and muscle contraction, might underlie the observed findings.

The MR results supported a connection between genetically proxied circulating levels of six cytokines and CD. Among them, associations of higher CXCL9/MIG and lower G-CSF/CSF-3 with CD were also supported by colocalization evidence. CXCL9/MIG, a proinflammatory IFNg-induced CXC-chemokine, is released by various immune cells, including macrophages, to attract and activate T-cells and forms together with neighboring CXCL10/IP-10, CXCL11/IP-9 and their cognate CXCR3 receptor an axis with specific relevance in inflammatory bowel disease[67,68]. In clinical studies, elevated CXCL9/MIG serum levels have been associated with CD relapses, affirming CXCL9/MIG as a risk factor in CD[55]. Apart from pro-inflammatory actions, exogenous administration of G-CSF/CSF-3 has been associated with immunoregulatory effects, such as modulation of T-cell responses[69]. Two open-label studies have indeed demonstrated that subcutaneous G-CSF/CSF-3 is effective in inducing clinical remission, fostering mucosal healing, and normalizing cell counts and cytokine responses in CD patients[56,57].

Integrating our results linking CD with circulating CXCL9/MIG levels, we identified genetically proxied expression of *TRAFD1* as a potential upstream causal regulator of CXCL9/MIG levels and, subsequently, risk of CD[55,70,71]. TRAFD1 binds as a homodimer or in interaction with TRAFD2, to TNF receptors, impacting pro-inflammatory cytokine production and modulating inflammatory responses in immune cells[34,35,72]. In celiac disease, *TRAFD1* was recognized as an upstream regulator of IFNg signaling and thereby activating cytotoxic T-cells, an important pathomechanism[36]. The identified function of *TRAFD1* as an inductor of IFNg signaling aligns well with the literature and also with our findings showing increased circulating CXCL9/MIG levels underlying *TRAFD1* expression[68]. Substantiating the risk-increasing associations in our analysis, elevated expression of *TRAFD1* and *TRAFD2* was noticed in acutely inflamed mucosal biopsies of CD patients[73]. Together these reports confirm our findings for the importance of *TRAFD1* in the pathology of CD and as a modulator of inflammatory reactions through effects on cytokine levels.

Our analysis also provided evidence for an inverse association between genetically proxied circulating TNF-b and MS risk. TNF-b is a pro-inflammatory cytokine within the TNF superfamily with substantial genetic correlation to TNF-a and binding affinity to pro-inflammatory as well as anti-inflammatory TNF receptors[74]. In a randomized phase 2 trial the TNF-inhibitor lenercept was tested for safety and efficacy in MS but had to be terminated prematurely after the interim analysis detected a dose-dependent increase in the frequency and severity of MS exacerbations[75]. In contrast to the TNF blockers infliximab, adalimumab and golimumab, lenercept provides equal inhibitory efficacy for TNF-a and TNF-b[74,76]. While TNF inhibition has demonstrated success in treating autoimmune diseases, such as rheumatoid arthritis or psoriasis, patients undergoing anti-TNF therapy for these indications are at risk for developing demyelinating CNS lesions, indicating a disease-specific effect[42,77]. Supporting the clinical findings, GWAS studies in MS identified TNF lowering alleles for both cytokines TNF-a and TNF-b that were associated with higher risk for MS[25,78]. We provide additional genetic evidence in line with observational, clinical, and GWAS findings for a potentially protective role of TNF-b in MS.

In our network analysis, over 80% of the significant cytokine–cytokine interactions led to an increase in downstream cytokine concentrations. Given that the majority of the involved cytokines were pro-inflammatory implies a self-perpetuating feedback mechanism leading to strong inflammatory responses and suggests a global trend where cytokines mutually activate each other to amplify their immune reaction. For the IL-1ra and TNF-b network, specifically, we successfully expanded the list of downstream affected cytokines. TNF-b exhibited notable downstream effects on 13 other cytokines, signifying its role as a master regulator. This observation aligns with existing reports on *TRAFD1's* downstream effects, influencing various cytokine-encoding genes like CXCL10/IP-10 and IFNg[36].

Our study has limitations. First, our meta-analysis was based on three cohorts that displayed considerable heterogeneity regarding the genetic background, the number of measured cytokines, and the biological specimen used across the databases. To objectify the influence of between database differences, we reported the numbers of variants with heterogenicity $p$-value < 0.1 and <0.05, respectively, and repeated the MR analyses based solely on variants without evidence of heterogeneity. The sensitivity analysis results are in line with our main findings showing pleiotropic downstream effects of TNF-b in the cross-cytokine MR and consistent cytokine–disease associations for asthma, CD, and MS. Nevertheless, the resulting heterogeneity (see Fig. 2) must be taken into consideration when interpreting the results. Next to these differences between the three cohorts, the databases also used different affinity-based assaying approaches for quantifying circulating cytokines. The different approaches might yield varying measurements for the same proteins with only moderate correlations across the assays[18]. This might explain the difference in replication rates across the three cohorts. Interestingly, we found a higher replication rate for signals detected with the Olink assay. In a previous cross-assay comparison between Olink and SomaScan, the proportion of assays with detected pQTLs was also higher with the Olink-based assay[18]. The

differences across the panels should be further explored at a larger scale to explore the extent to which it would be possible to scale genetic explorations across cohorts utilizing different proteomic platforms. Second, due to differences in reporting of effect sizes for genetic variants across the GWAS source data and the unavailability of individual-level data, we could perform only $p$-value-based meta-analyses and only indirectly estimate the pooled effect sizes based on the derived $p$-values and the variant allele frequencies. Inaccuracies in this estimation could influence downstream analyses heavily relying on effect sizes. Third, due to data availability, our analyses were based on 40 selected cytokines. Future endeavors utilizing solely high-throughput proteomic data could scale up to analyses including more inflammatory proteins. Fourth, our analyses are based on individuals of European, Finnish, and Icelandic ancestry and, as such, might not be generalizable to individuals of a different ancestry background. Fifth, due to the large number of cytokines, we adjusted the significance level for multiple testing, which might have neglected important findings due to non-significance. Using a hypothesis-driven approach, future studies should follow up on our results to identify additional targets we might have missed for a comprehensive view of our findings. Sixth our GWAS meta-analysis was based on population-based cohorts without predominant inflammatory diseases. Genetic variants might influence cytokine levels in specific contexts such as a response to infection or other pro-inflammatory stimuli; our approach could not detect such signals.

In conclusion, our study, leveraging data from 74,783 individuals across three cohorts, identified 169, mostly novel, genomic loci influencing circulating cytokine levels. Follow-up analyses of the detected signals reveal interesting underlying pathways, which enhance our understanding of the biology of the immune response. Integrating our data with genetic and clinical data for human disease risk, our analyses suggest potential targets like G-CSF/CSF-3, CXCL9/MIG, and TNF-b for immune-related diseases including asthma, Crohn's disease, and multiple sclerosis, warranting further exploration in clinical trials. The summary statistics from our study offer a valuable resource for future omics analyses, aiding data integration for the identification of potential drug targets for human diseases.

## Methods
### Study populations and design
The study cohorts and a flowchart of the study design are depicted in Fig. 1. We downloaded publicly available GWAS summary statistics for the circulating levels of up to 40 cytokines from three independent cohorts. We did not have access to individual-level data. Details of the study protocols have been published elsewhere[17,20,21]. Human genome assembly GRCh37 (hg19) from Genome Reference Consortium was used for genomic positioning[79]. Before further computations, all three databases were harmonized regarding data structure. For the GWAS meta-analyses and downstream computations, we included all cytokines that were available in at least two cohorts. To ensure that the available cytokines were identical between cohorts we used information provided on the NIH and Gene-Cards (https://www.genecards.org/) websites and verified synonyms and aliases in the abbreviated and full names of the cytokines. We excluded 1 cytokine (IL-12) from the analyses because different subunits of the protein were quantified in the two cohorts (IL-12p70 in YFS & FINRISK and IL-12p40 in deCODE).

**YFS and FINRISK**. Genomic data for 40 cytokines were drawn from up to 8293 individuals of Finnish background that were included in the YFS & FINRISK cohorts 1997 and 2002, respectively[17]. The mean age across all studies was 49 years (standard deviation 8 years). The cytokine measurements were carried out in EDTA plasma for the FINRISK 1997 cohort, in heparin plasma for the FINRISK 2002 cohort, and in serum for the YFS cohort using cytokine Luminex®-based multiplex immunoassays from Bio-Rad®. Genotyping was completed using the Illumina HT12 platform for the YFS study and the Illumina 670k HumanHap array for both FINRISK studies. Imputation was performed using the 1000 Genomes reference panel across all cohorts[79]. The GWAS meta-analyzing of

all three studies normalized the cytokine distribution using inverse transformation and adjusted the genetic analyses for age, sex, and ancestral principal components 1–10. The reported effect sizes were scaled per standard deviation increment in inverse-transformed cytokine levels.

**SCALLOP Consortium.** Genomic data for 16 cytokines were drawn from up to 30,931 individuals with European backgrounds from the SCALLOP consortium, a collaborative framework analyzing gene-protein associations across 13 studies[20]. The cytokine measurements were carried out in plasma samples using the proximity extension assay-based Olink® platform. Genotyping methods across the studies included Cardiometabochip, Immunochip, PsychChip, Illumina HumanCoreExome, Illumina OmniExpress, Metabochip, Illumina OmniExpress 2.5, Affymetrix Axiom UK Biobank array, HumanCytoSNP-12 BeadChip, HapMap300v2, Human Exome, Illumina HumanOmniExpressExome-8 v1, Illumina HumanHap300v1, Omni1, OmniX, Illumina Human-Hap300v1 and Infinium PsychArray-24 v1.2. Imputation was performed using the following panels: 1000G phase v5, 1000G phase v3, UK10K reference panel, HRC, HRC r1.1. The GWAS meta-analyzing all 13 studies adjusted the cytokines for age, gender, site, OLINK batch, Olink plate, MDS components, storage time, bleed to processing time (days), smoking status, oral contraceptive usage, blood cell counts, season of venipuncture and ancestral principal components 1–10. The log2-based normalized expression values (NPX) for each protein were rank-based inverse normal transformed and standardized to units of standard deviation.

**deCODE.** Genomic and proteomic data for 39 cytokines were taken from 35,559 Icelandic individuals included in deCODE[21]. The mean age was 55 years (standard deviation 18 years). The cytokine measurements were carried out in plasma samples using the aptamer-based SOMAScan® assay. Genotyping was completed using Illumina SNP Chip. Imputation was based on an in-house developed whole genome sequencing reference panel. The genetic analyses were adjusted for age and sex. Cytokine measurements were normalized using rank-inverse normal transformation and standardized to standard deviation increment. To allow alignment with other datasets, we excluded all SNPs that were not covered by the 1000 Genomes reference panel.

### Cross-assay comparisons in the genetic architecture of circulating cytokines

To explore differences in the genomic architecture of cytokines levels between the three studies that applied different measurement assays, we compared the proportion of overlapping SNPs between datasets confined to significant ($p$-value < 0.05) and directionally concordant (same direction of effect estimates across all three databases) variants. Using the GWAS summary level data of the three cohorts, we analyzed overlapping SNPs by taking one dataset as a reference and comparing it to the other two. Next, we calculated descriptive summary statistics (median and IQR) of the proportion of replicated SNPs across all cytokines for each cohort.

### GWAS meta-analyses

We performed fixed-effects inverse variance-weighted meta-analysis for each cytokine across the available cohorts using METAL software (v.2011-03-25, number of cohorts and sample sizes per cytokine GWAS are provided in Supplementary Data S8)[80]. Due to differences in scaling of the derived effect estimates across the three datasets, we applied a $z$-score-based meta-analysis (SCHEME SAMPLESIZE). Subsequently, we estimated standardized beta coefficients using $p$-values, minor allele frequency, and direction of effects, weighted according to sample sizes, as previously described[81]. For estimation of heterogeneity of effect sizes between the three genomic datasets (or two for cytokines available in only two of three datasets) we calculated Cochran $Q$ statistics for all included variants and re-run MR analyses (see Supplementary Figs. S2 and S3) excluding heterogenic

variants (HetPval < 0.05). To control for genomic inflation, we calculated lambda statistics for each cytokine (Supplementary Data S2)[82]. Significant variants were defined based on the established genome-wide significance level ($p < 5 \times 10^{-8}$). To detect independent variants following correction for linkage disequilibrium, we clumped across the significant ones using clump_data (TwoSampleMR R package version 0.5.6) at an $r^2 < 0.001$ based on the European 1000 Genomes Project reference panel[79]. We defined independent loci as SNPs that were separated by more than 1 Mb from the next SNPs in the 3' and 5' direction, as reported earlier[83].

### Linkage disequilibrium score regression (LDSC)

Using the LD score v1.0.1 tool we applied LDSC regression with reference data from the European 1000 Genomes project for calculation of cross-trait LDSC genetic correlations between all 40 cytokines using the meta-analysis results[79,84–86].

### Fine mapping, functional annotation, pathway, and gene-set analysis

To identify causal variants responsible for variations in circulating cytokine concentrations, we investigated significant loci associated with cytokines. We employed PLINK v1.9 to compute LD score correlation matrices and further refined the results using SuSiE (susieR R package version 0.12.16) to derive sets of variants, ensuring the inclusion of at least one causal variant with a cumulative probability ≥ 95%[87,88]. Subsequently, the causal variants were utilized to estimate the total variance explained by the identified loci for individual cytokines[89]. For functional analyses, we used phenoscanner (MendelianRandomization R package version 0.6.0) which ascribes functional consequences (intron, intergenic, exon, upstream, downstream, etc.) of single variants using positional mapping (physical distance)[90,91]. Gene-property analyses were conducted for identification of the tissue specificity of cytokines using the FUMA Gene2Func web database[92]. Lastly, MAGMA gene-based and gene-set analyses were conducted. Gene-based analysis initially calculates $p$-value association tests for variants mapped to protein-coding genes which are then used to calculate gene-set $p$-values in the gene-set analysis. Using predefined gene-sets, variants with significant associations to genes can then be analyzed to determine their underlying functional or process-related feature, i.e. gene-sets belonging to molecular functions or biological processes[93].

### Transcriptome-wide Mendelian randomization analysis (TWAS-MR)

To further explore whether variant effects on the expression of specific genes underlie the genetic underpinnings of circulating cytokine levels, we performed transcriptome-wide inverse variance-weighted 2-sample MR analysis, as has been previously described[94]. Sensitivity analyses were conducted using MR Egger regression and the weighted median estimator to control for horizontal pleiotropy[95,96]. For the calculation of effect estimates, we used the mr command from the TwoSampleMR R Package (TwoSampleMR version 0.5.6) with $cis$-expression quantitative trait loci (eQTL) gene instruments from the eQTLGen Consortium as exposure (clumped at $r^2 < 0.01$) and the GWAS meta-analysis results of our cytokine panel as outcome. Publicly available data was obtained from the eQTL consortium (summary level data accessible at https://eqtlgen.org/phase1.html) including 31,684 individuals of primarily European ancestry (detailed methods have been described previously and are available online)[24].

### Mendelian randomization analyses

We performed MR analyses exploring (i) the effects of circulating cytokine levels on other cytokines, (ii) the effects of circulating cytokine levels on allergic and autoimmune, cardiometabolic, and cancer disease endpoints, (iii) and, depending on the outcomes of the second MR analyses the effects of gene transcripts (eQTL) upstream of promising cytokines on allergic and autoimmune endpoints. For (i) and (ii) we used $cis$-acting variants as genetic instruments for our MR analyses, as they are associated with a lower risk of pleiotropic effects when compared to

*trans*-acting variants[97–101]. We filtered the GWAS meta-analysis results for variants within 300 kb around the gene encoding the respective cytokine. We selected variants associated at $p < 5 \times 10^{-5}$ and clumped the data at $r^2 < 0.1$. For the (i) cross-cytokine MR analyses, for each cytokine, we used as genetic instruments *cis*-variants influencing its levels and located within the locus of their encoding gene. This approach helped ensure that the variants were not directly used as instruments for two cytokines at the same time. Still, to exclude the possibility of pleiotropic mechanisms that might lead to spurious reverse causality associations being interpreted as causal, we also performed Steiger filtering excluding variants that showed stronger associations (larger absolute betas) with the "outcome" than the "exposure" cytokine. Steiger filtering led to the exclusion of 0.07% of instruments and replicated all of our main findings (see Supplementary Fig. S3 and Supplementary Data S6). For the (ii) disease endpoint MR we validated the significant MR results that also showed significant colocalization between cytokine levels and disease risk in the locus of the cytokine gene by using genetic instruments derived from an independent UKBB proteomics dataset, selected based on the same criteria. Furthermore, we explored whether measured levels of these cytokines are associated with the respective disease endpoints using clinical data from UKBB. The UKBB is a prospective cohort study that recruited over 500,000 individuals from the general UK population at baseline[102]. Between March 2006 and October 2010, participants aged 37–73 years attended one of 22 assessment centers across Scotland, England, and Wales[103,104]. Each participant completed a touchscreen questionnaire, had physical measurements taken, and provided blood, urine, and saliva samples at baseline. Plasma Proteomics in the UK Biobank dataset collected plasma samples from UKBB participants during their baseline visit. Samples were representative of the broader UK population, with 93% of European ancestry[105]. We performed logistic regression analyses adjusted for age and sex with the normalized levels of the proteins scaled at standard deviation increments. The outcomes were defined based on ICD-9 and ICD-10 codes of the primary care and hospital records of the study participants either before or after the baseline examination (Supplementary Data S8). To explore the discrimination utility of the protein biomarkers to detect or predict the disease endpoints, we calculated the area under the curve (c-index) for logistic regression models including age and sex, and models also including the protein levels. We applied fixed-effects inverse variance-weighted MR analysis as our main analytical approach[94]. Again, MR Egger regression and the weighted median estimator were used as sensitivity analyses[95,96]. After harmonization of the effect alleles across cytokines, we used the mr command from the TwoSampleMR R Package (TwoSampleMR version 0.5.6) to extract the respective effect estimates.

## Disease outcome GWASs

For the disease endpoints, we downloaded the largest, publicly available summary-level data GWAS based on European ancestry individuals that were non-overlapping with our cytokine summary-level data and performed MR analyses for three independent disease groups. For allergic and autoimmune phenotypes we analyzed asthma (121,940 cases, 1,254,131 controls)[106], Crohn's disease (5956 cases, 14,927 controls)[107], ulcerative colitis (6968 cases, 20,464 controls)[107], multiple sclerosis (47,429 cases, 68,374 controls)[108], psoriasis (4815 cases, 415,646 controls)[109], and rheumatoid arthritis (14,361 cases, 43,923 controls)[110]. For cardiometabolic phenotypes, we analyzed peripheral vascular disease (31,307 cases, 211,753 controls)[111], coronary artery disease (60,801 cases, 123,504 controls)[112], large artery stroke (9219 cases, 1,503,898 controls)[113] and diabetes mellitus type II (242,283 cases, 1,569,730 controls)[83]. For cancer phenotypes we analyzed breast cancer (133,384 cases, 113,789 controls)[114], colorectal cancer (5657 cases, 372,016 controls)[115], lung cancer (29,266 cases, 56,450 controls)[112], non-Hodgins lymphoma (2400 cases, 410,350 controls)[113], and skin cancer (23,694 cases, 372,016 controls)[115]. The data sources are detailed in Supplementary Data S8.

## Colocalization analysis

To analyze shared causal variants between SNPs for circulating cytokines and disease outcomes showing significant associations in MR analyses, we used the "coloc" v3 R package. COLOC is a variant colocalization method that performs tests on shared causal variants in the locus. Colocalization methods consider the GWAS and disease outcome summary statistics at a locus jointly and probabilistically test if the two signals are likely to be generated by the same causal variant[114]. We used the meta-analyses summary statistics for the significant cytokines restricted to a flanking region ±300 kb around the genetic location of each cytokine and mapped disease-associated variants by their rsID.

## Statistics and reproducibility

Baseline characteristics of each cohort were summarized in Supplementary Data S8. Meta-analysis of GWAS summary statistics was performed using METAL software (latest version released on 2011-3-25, https://csg.sph.umich.edu/abecasis/metal/download/), subsequent clumping, calculation of heterogeneity statistics and lambda was executed using R statistical environment with the help of R packages as defined in the code availability section. Next, we used LDSC v1.0.1 (https://github.com/bulik/ldsc) for the computation of LD Score Regression, heritability, and genetic correlation. To detect individual lead loci, we applied Plink 1.9 (version 1.9, https://www.coggenomics.org/plink/) to generate LD matrices that were incorporated by susieR R package to pinpoint causal variants. Functional analysis, including positional mapping, gene property, and gene-set analysis, was executed using the phenoscanner R package, the web-based tool FUMA (version 1.3.8 and 1.5.2, https://fuma.ctglab.nl/), and MAGMA (https://ctg.cncr.nl/software/magma, as implemented in FUMA), respectively. For all MR analyses in this manuscript, the TwoSampleMR R package was employed. Finally, colocalization was executed using the coloc R packages.

## Database search

To assess previously reported associations a database search was conducted using the NHGRI-GWAS catalog[115] on February 15, 2023. We analyzed our GWAS hits for associations with any of the 40 cytokines reported here (Supplementary Data S2), restricting the results for European-ancestry associations.

## Reporting summary

Further information on research design is available in the Nature Portfolio Reporting Summary linked to this article.

## Data availability

The data sources used in the current study are publicly available (download links are available in Supplementary Data S8). Ethical approval was not required due to the usage of publicly available summary-level data[17,20,21]. The datasets generated during and/or analyzed during the current study are available in the EBI GWAS catalog (accession numbers GCST90428399–GCST90428438). Whole-blood *cis*-eQTL summary statistics from the eQTLGen Consortium were downloaded from https://eqtlgen.org/phase1.html.

## Code availability

The analyses in this study were conducted using various line scripts as defined in the Statistics and Reproducibility section and R statistical environment (version 2024.09.1+394, https://www.r-project.org/). The R tools used in the analyses, which are publicly available, include: "TwoSampleMR" R package (version 0.5.6), "coloc" R package (version 5.2.3), MendelianRandomization R package (version 0.6.0), susieR R package (version 0.12.16). No custom algorithms or code were used in this study. An example R script can be found as a supplementary document named "code example.R". For further details on the analysis scripts or any specific code used in our study, please contact the corresponding author upon request.

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

## Acknowledgements

This work is supported by the German Research Foundation (Deutsche Forschungsgemeinschaft, DFG) by an Emmy Noether grant (GZ: GE 3461/2-1, ID 512461526 to M.G.K.), by a clinician-scientist grant to M.G.K. and project grants to J.B. and M.D. from the Munich Cluster for Systems Neurology (EXC 2145 SyNergy, ID 390857198), and by CRC1123 project A3 to J.B. Additional support comes from a research grant from the Fritz-Thyssen Foundation (Ref. 10.22.2.024MN to M.G.K.) and a research fellowship by the Hertie Foundation (Hertie Network of Excellence in Clinical Neuroscience, ID P1230035 to M.G.K.).

## Author contributions

M.J.K., M.O., L.Z., and M.K.G. conducted study-level analyses. T.G.R., R.M., S.E.B., J.B., M.D., and M.K.G. provided data and study supervision. M.J.K. and M.O. conducted the meta-analysis and downstream analyses. M.J.K., M.O., and M.K.G. drafted the manuscript. M.J.K. and M.K.G. conceived the project. All authors critically reviewed the manuscript and gave final approval to publish.

## Funding

## Competing interests

The authors declare the following competing interests: T.G.R. is an employee of GlaxoSmithKline unrelated to this work. M.K.G. has received consulting fees from Tourmaline Bio, Inc., unrelated to this work.
