## [Peer Review file · Communications Biology]

The Genomic Architecture of Circulating Cytokine Levels Points to Drug Targets for Immune-Related Diseases

Corresponding Author: Dr Marios Georgakis

This manuscript has been previously reviewed at another journal. This document only contains information relating to versions considered at Communications Biology.

Version 0:

Reviewer comments:

Reviewer #1

(Remarks to the Author)

The manuscript presents a rather comprehensive analysis of the genetic regulation of cytokine levels and their associations with several diseases. The line of thought of the manuscript is consistent and the results are of interest for researchers in the field. I particularly appreciate the methodology applied, as it uses several appropriate tools. I think the manuscript could be further improved, both in terms of presentation and results.

Major comments

- (1) In the last Results' section, the authors identify potential genes that act as mediators for disease. Validating these associations in independent datasets or databases (e.g. UK Biobank, DisGeNET) would add significant value. This could be done by measuring the performance of prediction models, based on the genes/cytokines identified by the authors, to predict individuals with diseases.
- (2) In the discussion, the authors note that a limitation of the study is the differences in how the GWAS analyses were performed. Since the authors have access to the raw data, why do they not perform the GWAS analyses themselves to ensure compatibility?
- (3) It is not clear how much of the cytokines' levels are regulated by genetics. The authors could add a multivariate analysis to show how much of the variance of each cytokine is influenced by genetics (all SNPs significantly associated to the cytokine) and other measured covariates (sex, age, etc.).
- (4) When performing MR analysis on cytokine-cytokine interactions, did the authors make sure to avoid SNPs that were associated with both cytokines (as it would go against the assumptions behind MR)?

Presentation

In my opinion, this is the part of the manuscript that could yield the most improvement.

- (1) In Fig. 1 I would consider adding more information regarding the compatibility between the three databases (e.g., Venn diagram on overlapping cytokine, or correlation between cytokine levels)
- (2a) Fig. 2a could be made more clear by ordering the bars from high to low.
- (2b) In Fig. 2b, I do not understand why the proportion of replication is accompanied by 25-75% quantiles; should it not be a single number?
- (2c) I would replace Fig. 2b with an UpSet plot.
- (3a) In Fig.4, the first letter of the axes' legends should be uppercase.
- (3b) I do not understand what the y-axis of Fig.4b represents
- (4) In Figure 5, since the same gene is associated to multiple cytokines, it could be useful to use a chord diagram to represent all the significant gene-cytokine associations.
- (5a) Why are there missing values in Fig. 7a?
- (5b) The authors could consider using "forceatlas" layout (e.g., <https://github.com/analyxcompany/ForceAtlas2>) to further emphasize which cytokines are similar to each other.
- (6) Is there a meaning behind the numbers on top of Figure 8?
- (7) I think some figures could be grouped together (1-2; 3-4; 5-6) to give a complete overview of the corresponding section.

Minor comments

- (1) Some technical terms, such "gene-based MAGMA", "gene-set MAGMA", "genetically proxied gene", are given for granted to the reader. Prefacing them with a very short description would facilitate the reading.
- (2) References should be inserted before the period that ends the sentence.
- (3) Please number the lines of the manuscript in future submissions (there is a number of typos).
- (4) Typo in the title of Supplementary Table 5.
- (5) The formatting of Supplementary Table 2 makes it impossible to read (possibly not the authors' fault).

Reviewer #2

(Remarks to the Author)

The three GWAS datasets used in this study have significant heterogeneity due to differences in genetic backgrounds (even though predominantly European, there is considerable genetic diversity), underlying measurement technologies, and the use of different biological samples (plasma and serum), etc. These factors can significantly contribute to the variability observed in the meta-analysis results, as shown in Figure 2. It's also noted that only 16 cytokines were included in the SCALLOP dataset, meaning that for the majority of traits, the analysis was based on only two studies. These all pose great challenges in addressing the underlying heterogeneity concern. The authors report that 48% of the identified loci have a heterogeneity p-value < 0.1 , but the rationale for selecting this threshold is unclear. It would be informative to know the percentage of traits with a heterogeneity p-value < 0.05 . Furthermore, while 156 novel loci are reported, it is not clear how many loci that were significant in the original studies lost significance after the meta-analysis. Overall, the current meta-analysis approach raises concerns about the reliability and trustworthiness of the results. A more carefully designed methodology is needed to select a proper set of high-quality results for downstream analysis.

Overall, there are several places in the manuscript that lack clarity or contain inaccuracies. For example, in Figure 2a, the method used to define replication of results is unclear and difficult to interpret. Are you reporting the number of significant SNPs per study and then categorizing them based on whether they were replicated in either of the other two studies? To enhance clarity, it would be helpful to directly provide the number of overlapping significant SNPs identified across the three datasets. Additionally, while Mendelian randomization (MR) analyses have been extensively conducted throughout the study, they lack details on, how the summary stats were properly selected, how confounders were controlled and does not adequately report the results of sensitivity analyses, etc. The omission of these details makes it challenging to fully trust the reported findings. Furthermore, there are multiple references to "cytokine serum levels" in the context of identified associations, but it should be noted that the meta-analyses were actually based on GWAS data from both serum and plasma.

Version 1:

Reviewer comments:

Reviewer #1

(Remarks to the Author)

The authors properly addressed my comments. I do not have further comments to add.

Reviewer #2

(Remarks to the Author)

I appreciate that the authors have addressed some of the concerns, but I believe there are still outstanding issues that need to be resolved.

1. The current sensitivity analysis, including or excluding high heterogeneity variants, clearly shows that the results are highly sensitive to these variants, i.e. when comparing Figures 7b and S2, as well as Figures 8a and S4. However, I'm not clear on how the authors concluded that "MR analyses of our main findings based solely on variants without evidence of heterogeneity confirm our main results."

Additionally, as I mentioned earlier, I would expect "a more carefully designed methodology." At present, there is no clear definition of how heterogeneity was determined and calculated, or how this was applied in a context where most traits are derived from only two studies. Does simply applying a p-value threshold sufficiently control for heterogeneity? I believe a post-analysis check would be worthwhile.

2. I also couldn't find Table S15, which was supposed to list the confounders that were adjusted for. Regarding the statement, "For the cross-cytokine MR and disease outcome MR, we have described the results of the sensitivity analyses in the Results section, and the full results are available in the supplement (Tables S5, S6, and S7)," I would suggest that sensitivity analyses need to be visualized; it is difficult to interpret just from numbers in the table.

In the MR analysis between cytokines, it is stated that "we used cis-acting variants as genetic instruments for our MR analyses." However, it's unclear how the cis variants overlap between cytokines/proteins. Do you mean that all cis variants

of all cytokines were used?

In the main text, it states, "Using cis-acting variants for G-CSF/CSF-3, CXCL9/MIG, and TNF-b from the UKBB proteomics dataset as instruments, we confirmed significant associations with asthma, Crohn's disease, and multiple sclerosis, respectively." Are these the only three proteins from UKBB that overlap with the 40 included in this study, and did they replicate all the associations identified in the MR with UKBB data? This context is crucial for interpreting the results.

3. There are also other potential errors, such as in the phrase "minimum allele frequency," which I believe should be "minor allele frequency."

4. Finally, Supplementary Table S8 only provides data sources for the 40 cytokines but does not include those for the diseases used in the MR analysis.

Version 2:

Reviewer comments:

Reviewer #2

(Remarks to the Author)

The authors have addressed my questions. I don't have anything further to add.

Reviewer #1 (Remarks to the Author):

The manuscript presents a rather comprehensive analysis of the genetic regulation of cytokine levels and their associations with several diseases. The line of thought of the manuscript is consistent and the results are of interest for researchers in the field. I particularly appreciate the methodology applied, as it uses several appropriate tools. I think the manuscript could be further improved, both in terms of presentation and results.

We thank the reviewer for critically appraising our work and for providing thoughtful comments, which we address point-by-point below.

Major comments

(1) In the last Results' section, the authors identify potential genes that act as mediators for disease. Validating these associations in independent datasets or databases (e.g. UK Biobank, DisGeNET) would add significant value. This could be done by measuring the performance of prediction models, based on the genes/cytokines identified by the authors, to predict individuals with diseases.

We thank the reviewer for the constructive feedback. To address the concerns, we performed additional analyses making use of the independent UK Biobank cohort. The results of this analysis are presented in the updated Figure 8b, which is also attached below:

First, we validated the top 4 MR results (that also showed significant colocalization) using GWAS data for proteins quantified within the Olink proteomics assay in the UK Biobank. The figure on the left illustrates the results of MR-IVW analysis using instruments from our own meta-analysis GWAS data and from the independent UK Biobank data – the results were highly consistent, thereby validating our conclusions.

Second, as recommended by the reviewer, we assessed whether the levels of the quantified proteins in the UK Biobank were associated with the respective disease endpoints, as recorded in ICD-9 and ICD-10 diagnoses in the UK Biobank. As illustrated on the figure on the right, circulating cytokine levels for all 4 cytokine-disease associations predicted disease risk in a logistic regression model adjusted for age and sex with 3 of the associations being directionally consistent to the MR results. The c-index of an age- and sex-adjusted model for asthma and multiple sclerosis did not significantly improve after adding G-CSF (0.558 to 0.562) and TNF-b levels (0.612 to 0.618), respectively, but the addition of MIG levels led to significant prediction gains for Crohn's disease (0.536 to 0.659).

	Overall C_index (95% CI)	C_index changes
Asthma (n=3646)		
Age+Sex	0.547 (0.537-0.557)	Reference
Plus G_CSF	0.547 (0.537-0.557)	0
Crohn's disease (n=189)		
Age+Sex	0.561 (0.520-0.602)	Reference
Plus G_CSF	0.593 (0.550-0.636)	0.032
Plus MIG	0.642 (0.600-0.683)	0.081
Multiple sclerosis (n=230)		
Age+Sex	0.614 (0.579-0.650)	Reference
Plus TNFb	0.619 (0.584-0.654)	0.005

We integrated these methods and results in the manuscript, as follows:

-Methods (**Page 22, Line 1057ff.**): „For the (ii) disease endpoint MR we validated the results using genetic instruments derived from an independent United Kingdom Biobank (UKBB) proteomics dataset, selected based on the same criteria. Furthermore, we explored whether measured levels of these cytokines are associated with the respective disease endpoints using clinical data from UKBB. The UKBB is a prospective cohort study that recruited over 500,000 individuals from the general UK population at baseline 98. Between March 2006 and October 2010, participants aged 37–73 years attended one of 22 assessment centers across Scotland, England, and Wales 99,100. Each participant completed a touchscreen questionnaire, had physical measurements taken, and provided blood, urine, and saliva samples at baseline. Plasma Proteomics in the UK Biobank dataset collected plasma sample from UKB participants during their baseline visit. Samples were representative of the broader UKB population, with 93% of European ancestry.¹⁰¹ We performed logistic regression analyses adjusted for age and sex with the normalized levels of the proteins scaled at standard deviation increments. The outcomes were defined based on ICD-9 and ICD-10 codes of the primary care and hospital records of the study participants either before or after the baseline examination (Supplementary Table S8). To explore the discrimination utility of the protein biomarkers to detect or predict the disease endpoints, we calculated the area under the curve (c-index) for logistic regression models including age and sex, and models also including the protein levels.“

-Results (**Page 9, Line 393ff.**): „To substantiate our MR results we validated our findings with genetic instruments derived from the independent UKBB data. Using cis-acting variants for G-CSF/CSF-3, CXCL9/MIG and TNF-b from the UKBB proteomics dataset as instruments we could confirm significant associations with asthma, CD and MS, respectively. Furthermore, we performed logistic regression analyses for these cytokines and the respective disease endpoints defined by ICD-9 and ICD-10 codes in the UKBB with 3 of the associations being significant and directionally consistent to the MR results (Figure 8b). The c-index of an age- and sex-adjusted model for asthma and multiple sclerosis did not significantly improve after adding G-CSF (0.558 to 0.562) and TNF-b levels (0.612 to 0.618), respectively, but the addition of MIG levels led to significant prediction gains for Crohn’s disease (0.536 to 0.659).“

(2) In the discussion, the authors note that a limitation of the study is the differences in how the GWAS analyses were performed. Since the authors have access to the raw data, why do they not perform the GWAS analyses themselves to ensure compatibility?

We thank the reviewer for raising this important point. Indeed, as we note in the discussion, the differences in how GWAS analyses were performed is a key limitation of our work and the results should be interpreted accordingly. Unfortunately, we do not have access to individual-level data of any of the three studies, which would allow to reperform these analyses. We apologize if this was not clear

in the original version of the manuscript. To address the issue and avoid lack of clarity, we added a sentence to the methods highlighting the lack of access to individual-level data (**Page 20, Line 921f.**): *“We downloaded publicly available GWAS summary statistics for the circulating levels of up to 40 cytokines from 3 independent cohorts. We did not have access to individual-level data.”*

(3) It is not clear how much of the cytokines' levels are regulated by genetics. The authors could add a multivariate analysis to show how much of the variance of each cytokine is influenced by genetics (all SNPs significantly associated to the cytokine) and other measured covariates (sex, age, etc.).

We thank the reviewer for the suggestion. We have calculated the variance explained by the significant SNPs for each cytokine and this is provided in Supplementary Table 2, as well as in the Results (**Page 5, Line 167f.**): *“The proportion of explained variance by the significant variants ranged from 0.0008 for IL-17 to 0.033 for stem cell growth factor beta (SCGF-b) (Supplementary Table S2).”*

We hope that this answers the reviewer's comment. Unfortunately, it was not possible to perform multivariate analysis, as recommended by the reviewer, because we lacked access to individual-level data of the original three GWASs.

(4) When performing MR analysis on cytokine-cytokine interactions, did the authors make sure to avoid SNPs that were associated with both cytokines (as it would go against the assumptions behind MR)?

We apologize for not being sufficiently clear in our manuscript. Performing MR analysis on cytokine-cytokine interactions we used *cis*-acting variants as instruments. Thereby we precluded the possibility that the same variants were used for both the exposure and the outcome, thus conforming with MR assumptions. Using *cis*-variants, we minimize the probability that the effects of a variant on cytokine levels are due to pleiotropic effects through another gene that could also influence another cytokine. Still, there might be bias if the variants exert their effects through neighboring genes – to address this possibility, we did Steiger filtering excluding any variants that were more strongly associated with the “outcome” than the “exposure” cytokine. This approach led to exclusion of only a minor portion (0.07 %) of instruments and replicated all of our main findings.

We now aimed to better clarify in our methods the procedure for performing MR analysis on cytokine-cytokine interactions (**Page 22, Line 1047ff.**): *“For (i) and (ii) we used cis-acting variants as genetic instruments for our MR analyses, as they are associated with a lower risk of pleiotropic effects when compared to trans-acting variants. We filtered the GWAS meta-analysis results for variants within 300 kb around the gene encoding the respective cytokine. We selected variants associated at $p < 5 \times 10^{-5}$ and clumped the data at $r^2 < 0.1$. For the (i) cross-cytokine MR analyses, using cis-acting variants helped ensure that the variants were not directly used as instruments for two cytokines at the same time. Still, to exclude the possibility of pleiotropic mechanisms that might lead to spurious reverse causality associations being interpreted as causal, we also performed Steiger filtering excluding variants that showed stronger associations (larger absolute betas) with the “outcome” than the “exposure” cytokine. Steiger filtering led to exclusion of 0.07% of instruments and replicated all of our main findings (see Supplementary Figure S3 and Supplementary Table S6).”*

Presentation

In my opinion, this is the part of the manuscript that could yield the most improvement.

We thank the reviewer for pointing out presentation aspects of our work that can be improved and giving detailed suggestions for better visualization of our data.

(1) In Fig. 1 I would consider adding more information regarding the compatibility between the three databases (e.g., Venn diagram on overlapping cytokine, or correlation between cytokine levels).

We thank the Reviewer for this suggestion. We updated Figure 1 adding a Venn diagram as panel B.

(2a) Fig. 2a could be made more clear by ordering the bars from high to low.

As recommended, we ordered the bars in Figure 2a from high to low with respect to the total number (across databases) of replicated SNPs, meaning CCL4/MIP-1b has the highest number of replicated SNPs (saturated part of the bars) when adding up SNPs_{replicated} (SCALLOP) and SNPs_{replicated} (YFS & FINRISK).

(2b) In Fig. 2b, I do not understand why the proportion of replication is accompanied by 25-75% quantiles; should it not be a single number?

Figure 2b represents the median replication across all cytokine GWASs. As such, we included the IQR (25th to 75th quantile) as a measure of variability of the proportion of hits replicated for each cytokine. We adapted the Figure legend hoping that this is now clarified (**Page 4, Line 135ff.**): “Figure 2. Comparisons of significant genomic loci for 40 circulating cytokines across 3 proteomics assays. (a) Number of reproducible and non-reproducible SNPs per cytokine (depicted as saturated and light-colored bars, respectively) for deCODE, SCALLOP and YFS & FINRISK cohorts. The saturated portion of the bars represents the number of SNPs that were replicated in at least one other cohort, where reproducibility is defined as SNPs confined to significant loci (p -value < 0.05) and directionally concordant. (b) Proportion of replicated SNPs across cytokine datasets visualized for all possible combinations of cohorts. The matrix at the bottom left shows the comparison each of the vertical bars at the top represents. Arrows in the comparison matrix illustrate the direction of comparison—from the reference dataset where significant SNPs were identified, to the dataset in which SNPs were replicated. For example, the first bar displays the percentage of loci in deCODE that were replicated in SCALLOP (with the arrow pointing from deCODE to SCALLOP). The horizontal bars on the bottom right shows the number of cytokines for which significant SNPs were found. Median, IQR (error bars represent the 25th and 75th percentiles). Colored bars represent deCODE consortium in red, SCALLOP consortium in blue and YFS & FINRISK cohorts in yellow. IQR, inter-quartile range. SNP, Single-nucleotide polymorphism; SCALLOP, Systematic and Combined Analysis of Olink Proteins; YFS & FINRISK, Cardiovascular Risk in Young Finns Study.)”

(2c) I would replace Fig. 2b with an UpSet plot.

Although our meta-data are more high-dimensional than the data usually presented in an UpSet plot, we reformatted the figure to fit with an UpSet framework.

(3a) In Fig.4, the first letter of the axes' legends should be uppercase.

As recommended, an updated version of Figure 4 has been provided with correction of the typos.

(3b) I do not understand what the y-axis of Fig.4b represents

The Y-axis represents the number of significant loci, binned by the amount of cytokine associated with each locus. We reframed the Figure legend for better understanding (**Page 5, Line 198f.**): “Number of significant loci binned by the number of associated circulating cytokines (excluding the HLA region on chromosome 6).”

(4) In Figure 5, since the same gene is associated to multiple cytokines, it could be useful to use a chord diagram to represent all the significant gene-cytokine associations.

As suggested, we added a chord diagram as Figure 5b.

(5a) Why are there missing values in Fig. 7a?

We thank the reviewer for pointing this out and apologize for not clarifying this in the first version of the manuscript. Missing values arose due to a failure of the LDSC to compute correlations due to the extremely low SNP heritability for two cytokines (MIF and G-CSF/CSF-3) leading to negative h^2 values, as computed with the LDSC method. We added the explanation to the results (Page 7, Line 312ff.): “Due to a computation error in 5% of cross-cytokine interactions (80 of 1600) caused by missing evidence of SNP-heritability for 2 phenotypes (MIF [Macrophage migration inhibitory factor] and G-CSF/CSF-3), LDSC values and therefore correlations could not be computed.”

And added a note to the figure legend of Figure 7a (Page 8, Line 344ff.): “Figure 7. Cross-cytokine genetic associations. (a) Genetic correlations with LD-score regression across cytokine serum levels are depicted as correlation heatmap. Stars highlight significance level *, 0.05; **, 0.0001; ***, 0.00001. LD-score correlation coefficients are illustrated according to the legend below spanning from -1 in blue to +1 in red, missing correlation coefficients are depicted in grey and are due to no evidence of SNP-heritability for one of the cytokines.”

(5b) The authors could consider using "forceatlas" layout (e.g., <https://github.com/analyxcompany/ForceAtlas2>) to further emphasize which cytokines are similar to each other.

We appreciate the reviewer's suggestions for visual improvement of our work. However, we believe that a forceAtlas layout would not align with the information we intend to convey. In this figure we want to emphasize on "cytokines-hubs" that are tightly interconnected and exert top-down regulatory functions on other cytokines and not so much on similarities between cytokines. Additionally, we aim to convey directionality of the effects, as well as magnitude of the effect sizes, which would not be possible to illustrate in a forceatlas layout.

(6) Is there a meaning behind the numbers on top of Figure 8?

We thank the reviewer for bringing up this point, as it was not well communicated in the manuscript or figure legend. In Figure 8 the numbers atop represent the number of *cis*-acting genetic variants used as instruments in the MR analyses. An updated version of Figure 8 has been provided and the figure legend has been updated (**Page 9, Line 410f.**): "*n(SNP)*, indicates number of *cis*-acting genetic variants used as instruments in MR analyses".

(7) I think some figures could be grouped together (1-2; 3-4; 5-6) to give a complete overview of the corresponding section.

Due to the large number of multi-panel or data-rich figures, we find it hard to combine them at their current format. If the editors or the reviewer feel strongly that this is necessary to improve the communication of our results, we would be happy to reconsider.

Minor comments

(1) Some technical terms, such "gene-based MAGMA", "gene-set MAGMA", "genetically proxied gene", are given for granted to the reader. Prefacing them with a very short description would facilitate the reading.

We thank the reviewer for pointing this out. We added a concise explanation shortly summarizing the methodological terms in the results sections next to the explanation that was already in place in the methods section:

Page 6, Line 218ff.: "*We performed a gene-based MAGMA (Multi-marker Analysis of GenoMic Annotation) analysis, which combines the effects of multiple SNPs to identify associations between genes and circulating cytokine levels.*"

and **Page 6, Line 233ff.:** "*Further combining the genes to sets related to concrete biological pathways, we performed a gene-set MAGMA analysis which prioritized 41 pathways for 12 cytokines that reached a Bonferroni-adjusted significance level.*"

(2) References should be inserted before the period that ends the sentence.

As recommended, we adapted the references accordingly.

(3) Please number the lines of the manuscript in future submissions (there is a number of typos).

We numbered the lines as recommended by the reviewer.

(4) Typo in the title of Supplementary Table 5.

We have corrected the typo in the title of Supplementary Table 5 and screened the manuscript for additional typos.

(5) The formatting of Supplementary Table 2 makes it impossible to read (possibly not the authors' fault).

We thank the reviewer for the detailed observation. We have uploaded our supplementary tables as excel files, which is how we would also like to have them uploaded in case the manuscript gets accepted. The formatting of all Supplementary Tables will be thoroughly reviewed before uploading the manuscript to the submission portal. We hope that it will be possible for you to download the original excel files for review.

Reviewer #2 (Remarks to the Author):

The three GWAS datasets used in this study have significant heterogeneity due to differences in genetic backgrounds (even though predominantly European, there is considerable genetic diversity), underlying measurement technologies, and the use of different biological samples (plasma and serum), etc. These factors can significantly contribute to the variability observed in the meta-analysis results, as showed in Figure 2. It's also noted that only 16 cytokines were included in the SCALLOP dataset, meaning that for the majority of traits, the analysis was based on only two studies. These all pose great challenges in addressing the underlying heterogeneity concern. The authors report that 48% of the identified loci have a heterogeneity p-value < 0.1, but the rationale for selecting this threshold is unclear. It would be informative to know the percentage of traits with a heterogeneity p-value < 0.05.

We thank the reviewer for the very constructive feedback and for raising this important point regarding the various sources of heterogeneity in our analysis.

To address the reviewer's comment about heterogeneity, we now report the percentage of variants with a heterogeneity p-value < 0.05, which is 41%, and present a new table in the supplement with all of these variants (new Supplementary Table 2). As many of these signals might be driven by strong results in one of the datasets, to strengthen the credibility of our downstream analyses, we re-ran the MR analyses of cross-cytokine and cytokine-disease associations after excluding variants with a heterogeneity p-value < 0.05 (please see supplementary Figures S2 and S4). Using variants without evidence of heterogeneity as instruments in MR analyses, we showed significant and directionally consistent results for 3 out of 6 cytokines that were positively associated with > 2 other cytokines in the cross-cytokine MR analysis. Moreover, we validated our main finding showing pleiotropic downstream effects of TNF- β inducing higher circulating levels on 12 (out of 13) pro-inflammatory cytokines. In MR analysis on disease associations the re-ran analysis replicated 12 out of 24 significant associations including the colocalized results obtained for Asthma, CD and MS.

We reported our findings in result section (**Page 4, Line 157ff.**): *"Variants that showed significant heterogeneity between the three cohorts (HetPval < 0.1) are reported in Supplementary Table S2 (48% of the significant loci with HetPval < 0.1; range 0% to 100% across 40 cytokines and 41% of the significant loci with HetPval < 0.05; range 0% to 100% across 31 cytokines)." and referred to the results of the new MR analyses in the respective paragraphs.*

For the cross-cytokine MR analysis (**Page 7, Line 316ff.**): *"We found significant (FDR-corrected $p < 0.05$) associations between 65 cytokine pairs (53 positive associations and 12 negative associations, Figure 7b, sensitivity analyses excluding variants showing significant heterogeneity between the three datasets in Supplementary Figure S2 and Supplementary Table S6)." and referred to the results of the new MR analyses in the respective paragraphs.*

and for the disease association MR analysis (**Page 8, Line 358ff.**): *"For insights into the clinical consequences of genetically proxied levels of the circulating cytokines, we analyzed associations with allergic and autoimmune, cardiometabolic, and cancer outcomes in two-sample MR followed by colocalization analyses (Figure 8a, sensitivity analyses excluding variants showing significant heterogeneity between the three datasets in Supplementary Figure S4 and Supplementary Table S7)." and referred to the results of the new MR analyses in the respective paragraphs.*

Furthermore, we now more explicitly highlight in the limitations of this work, the issues raised by the reviewer, aiming to attract the readers' attention to heterogeneity issues that need to be considered when interpreting the results of our work. Please see **page 12, line 536ff.**: *"Our study has limitations. First, our meta-analysis was based on 3 cohorts that displayed considerable heterogeneity regarding the genetic background, the number of measured cytokines, and the biological specimen used across the databases. To objectify the influence of between database differences we reported the numbers of variants with heterogeneity P-value < 0.1 and < 0.05, respectively and repeated the MR analyses of our*

main findings based solely on variants without evidence of heterogeneity, confirming our main results. Nevertheless, the resulting heterogeneity (see Figure 2) must be taken into consideration when interpreting the results. Next to these differences between the 3 cohorts, the databases also used different affinity-based assaying approaches for quantifying circulating cytokines. The different approaches might yield varying measurements for the same proteins with only moderate correlations across the assays 18. This might explain the difference in replication rates across the 3 cohorts. Interestingly, we found a higher replication rate for signals detected with the Olink assay. In a previous cross-assay comparison between Olink and SomaScan, the proportion of assays with detected pQTLs was also higher with the Olink-based assay 18. The differences across the panels should be further explored at a larger scale to explore the extent to which it would be possible to scale genetic explorations across cohorts utilizing different proteomic platforms.”

Furthermore, while 156 novel loci are reported, it is not clear how many loci that were significant in the original studies lost significance after the meta-analysis.

We thank the reviewer for pointing out this important aspect of our analysis. To address this point, we now present the number of loci that became non-significant after meta-analysis (across databases = 17,6%), reported our results in supplementary Table S2 and added them to the main text (**Page 5, Line 165f.**): *“Assessing the loss of loci that were significant in the original studies, we found 18% of loci lost significance in the meta-analyzed samples of the 3 source databases”.*

Overall, the current meta-analysis approach raises concerns about the reliability and trustworthiness of the results. A more carefully designed methodology is needed to select a proper set of high-quality results for downstream analysis. Overall, there are several places in the manuscript that lack clarity or contain inaccuracies. For example, in Figure 2a, the method used to define replication of results is unclear and difficult to interpret. Are you reporting the number of significant SNPs per study and then categorizing them based on whether they were replicated in either of the other two studies? To enhance clarity, it would be helpful to directly provide the number of overlapping significant SNPs identified across the three datasets.

We thank the reviewer for the direct feedback and the opportunity to address this important point in our work. Indeed, we are reporting the number of significant SNPs per study and then categorizing them based on whether they were replicated in either of the other two studies. Apparently, this methodological aspect was not clearly communicated. Furthermore, as recommended in the first comment by the reviewer, we have now run alternative versions of our downstream MR analyses restricted to a set of genetic variants that showed no heterogeneity across the three datasets.

To address the reviewer's comment about Figure 2, we have changed the methods and results sections as well as the visualization of Figures 2a and 2b to enhance clarity:

-Methods (Page 21, Line 978ff.): *“To explore differences in the genomic architecture of cytokines levels between the 3 studies that applied different measurement assays, we compared the proportion of overlapping SNPs between datasets confined to significant (p -value < 0.05) and directionally concordant (same direction of effect estimates across all 3 databases) variants. Using the GWAS summary level data of the 3 cohorts, we analyzed overlapping SNPs by taking one dataset as reference comparing it to the other two. Next, we calculated descriptive summary statistics (median and IQR) of the proportion of replicated SNPs across all cytokines for each cohort.”*

-Results (Page 4, Line 119ff.): *“Although the GWAS in SCALLOP identified a considerably lower number of genome-wide significant loci for the available cytokines ($n=119$ SNPs found for 13 cytokines), these variants exhibited the highest reproducibility rate ($p<0.05$ and directionally consistent) across the other two datasets. Specifically, 79 out of 119 variants replicated in YFS and FINRISK, with a median replication rate of 67%, and 46 out of 77 variants replicated in deCODE, with a median replication rate*

of 63% (Figure 2, Supplementary Figure S1, and Supplementary Table S1). In contrast, variants identified as significant in YFS & FINRISK had a lower replication rate in deCODE (70 out of 785 variants replicated, with a median replication rate of 4%) and in SCALLOP (101 out of 481 variants replicated, with a median replication rate of 11%). Similarly, variants identified in deCODE showed relatively low reproducibility with a median replication rate of 21% in YFS & FINRISK (141 out of 607 SNPs replicated) and 19% in SCALLOP (73 SNPs out of 252 replicated)."

Additionally, while Mendelian randomization (MR) analyses have been extensively conducted throughout the study, they lack details on, how the summary stats were properly selected, how confounders were controlled and does not adequately report the results of sensitivity analyses, etc.

We thank the reviewer for pointing out this issue and we apologize if the MR methods and results lacked clarity. To address the comment, we implemented the following changes in the Methods section of the manuscript:

(i) we added a description on the criteria used to select the summary stats (**Page 23, Line 1082ff.**): *"For the disease endpoints, we downloaded the largest, publicly available summary level data GWAS's based on European ancestry individuals that were non-overlapping with our cytokine summary level data and performed MR analyses for 3 independent disease groups "*

(ii) we added information about the confounders that were adjusted for in the single disease databases in Table S15.

(iii) For the cross-cytokine MR, and disease outcome MR we have described the results of the sensitivity analyses in the results section and made the full results available in the supplement (Tables S5, S6 and S7):

- cross-cytokine MR (**Page 7, Line 320ff.**): *"In pleiotropy-robust sensitivity analyses (weighted median MR, MR Egger) 89% and 85% of the 65 cytokine pairs showed directionally concordant associations, respectively (Table S6)."*

- cross-cytokine MR (**Page 8, Line 336ff.**): *"For both master regulator cytokines (i.e. TNF-b and IL-1ra) sensitivity analyses confirmed directional consistency for all associations using the two MR methods reported above (Supplementary Table S6)."*

- disease outcome MR (**Page 8, Line 368ff.**): *"Sensitivity analyses showed directional concordance for 91% and 75% of the associations calculated with weighted median MR and MR Egger, respectively (Supplementary Table S7)."*

The omission of these details makes it challenging to fully trust the reported findings. Furthermore, there are multiple references to "cytokine serum levels" in the context of identified associations, but it should be noted that the meta-analyses were actually based on GWAS data from both serum and plasma.

We thank the Reviewer for this comment. Accordingly, to avoid any misconception regarding the source of cytokine measurements, we added an explanation on the wording used throughout the manuscript in the introduction (**Page 3, Line 63ff.**): *"Circulating cytokines (i.e. cytokines measured in the circulation including serum and blood specimen) are readily [...]"*. We have made sure to avoid the term "cytokine serum levels" throughout.

Reviewers' comments:

Reviewer #2 (Remarks to the Author):

I appreciate that the authors have addressed some of the concerns, but I believe there are still outstanding issues that need to be resolved.

We thank the reviewer for critically appraising our work and for providing thoughtful comments, which we address point-by-point below.

1. The current sensitivity analysis, including or excluding high heterogeneity variants, clearly shows that the results are highly sensitive to these variants, i.e. when comparing Figures 7b and S2, as well as Figures 8a and S4. However, I'm not clear on how the authors concluded that "MR analyses of our main findings based solely on variants without evidence of heterogeneity confirm our main results."

We thank the reviewer for raising this important point. Indeed, as noted by the reviewer the heterogeneity in the dataset led to differences in the overall MR results for the cross-cytokine analysis (Figures 7b and S2) as well as for the drug-target analysis (Figures 8a and S4). Nevertheless, based on the highly consistent results in the sensitivity analyses regarding the network interactions of TNF-b and the cytokine-disease associations for asthma, CD and MS, which constitute our most important findings, we concluded that our main results were replicated. To preclude misunderstandings, we reframed the sentence pinpointing the specific results that were replicated in the sensitivity analyses (**Page 12, Line 541ff**): *"To objectify the influence of between database differences we reported the numbers of variants with heterogeneity P-value < 0.1 and < 0.05, respectively and repeated the MR analyses based solely on variants without evidence of heterogeneity. The sensitivity analyses results are in line with our main findings showing pleiotropic downstream effects of TNF-b in the cross-cytokine MR and consistent cytokine-disease associations for asthma, CD and MS."*

Additionally, as I mentioned earlier, I would expect "a more carefully designed methodology." At present, there is no clear definition of how heterogeneity was determined and calculated, or how this was applied in a context where most traits are derived from only two studies. Does simply applying a p-value threshold sufficiently control for heterogeneity? I believe a post-analysis check would be worthwhile.

We thank the Reviewer for this comment about the important issue of heterogeneity. We have until now undertaken the following actions to consider heterogeneity in the interpretation of our findings:

- Firstly, to explore differences in the genomic architecture of cytokine levels between the 3 studies that applied different measurement assays, we assessed the proportion of overlapping SNPs between them (these results are visualized in our results in **Figure 2**). As pointed out by the reviewer in the first review, the methodology and findings from the cross-assay comparison were difficult to understand. Consequently, we worked on the methods and results sections as well as on the visualization to enhance clarity and interpretability.

- Secondly, for quantifying heterogeneity of effect sizes between data sources we calculated the Cochran Q statistics and reported variants with HetPval < 0.1 and additionally, as requested by the reviewer, also with HetPval < 0.05. We now clarify in the methods that we used the Cochran Q statistic for assessing heterogeneity, even between two datasets, as follows (**Page 22, Line 1043ff**):

"For estimation of heterogeneity of effect sizes between the three genomic datasets (or two for cytokines available in only two of three datasets) we calculated Cochran Q statistics for all included

variants and re-run MR analyses (see Supplementary Figure S2 and S3) excluding heterogenic variants ($HetPval < 0.05$).”

Finally, as the reviewer recommends, we have conducted post-hoc validation of our findings by excluding heterogenic variants and calculated cross-cytokine and cytokine-disease MR analyses. The analyses replicated our main findings showing pleiotropic downstream effects of TNF-b in the cross-cytokine MR and consistent cytokine-disease associations for asthma, CD and MS.

2. I also couldn't find Table S15, which was supposed to list the confounders that were adjusted for. Regarding the statement, "For the cross-cytokine MR and disease outcome MR, we have described the results of the sensitivity analyses in the Results section, and the full results are available in the supplement (Tables S5, S6, and S7)," I would suggest that sensitivity analyses need to be visualized; it is difficult to interpret just from numbers in the table.

We apologize for the typing error in the rebuttal letter. The sentence should read: "(ii) we added information about the confounders that were adjusted for in the single disease databases in **Table S8**". Table S8 shows the following overview of the disease databases including confounders adjusted for:

Supplementary Table S8: characteristics of the included disease outcome

disease group	phenotype	Total N	Cases n	Control n	Study PMID / Author, Year	Population	source download link	Confounder adjustment
cardiometabolic	Peripheral vascular disease	243060	31307	211753	31285632 / Klarin et al., 2019	European	https://www.ncbi.nlm.nih.gov/projects/gap/cgi-bin/study.cgi	age, sex, and 5 PCs
	Coronary artery disease	184305	60801	123504	26343387 / Nikpay et al., 2015	European	www.ccardiogramplus4d.org	ethnic outliers, related individuals, heterozygosity, gender mismatch, related individuals and duplicates, stenosis $\geq 10\%$ and $\leq 50\%$
	Largery artery stroke	1513117	9219	1503898	36180795 / Mishra et al., 2022	European	http://www.megastroke.org/	age, sex, PCs of population stratification and study-specific covariates when needed
cancer	diabetes mellitus type II	1812013	242283	1569730	38374256 / Suzuki et al. 2024	European	http://www.diagram-consortium.org/downloads.html	population structure and relatedness, age and sex
	breast	247173	133384	113789	32424353 / Zhang et al. 2020	European	https://github.com/Wittelab/pancancer_pleiotropy	PCs, age
	colorectal	377673	5657	372016	10.5523/BRIS.AED0U12W0EDE200	European	https://data.bris.ac.uk/data/dataset/aed0u12w0ede200lb0m	Genotype array and sex
	lung	85716	29266	56450	28604730 / McKay et al. 2017	European	https://www.ncbi.nlm.nih.gov/projects/gap/cgi-bin/study.cgi	age, sex and genetically derived ancestry
	non-Hodgkins lymphoma	412750	2400	410350	32887889 / Rashkin et al. 2020	European	https://github.com/Wittelab/pancancer_pleiotropy	age at specimen collection, sex, first ten ancestry PCs, genotyping array (UKB only), and reagent kit used for genotyping
skin	395710	23694	372016	10.5523/BRIS.AED0U12W0EDE200	European	https://data.bris.ac.uk/data/dataset/aed0u12w0ede200lb0m	Genotype array and sex	
allergic and autoimmune	asthma	1376071	121940	1254131	36778051 / Tsuo et al., 2022	European	http://results.globalbiobankmeta.org/	age, age ² , sex, age*sex, 20 first PCs, and any biobank specific covariates
	crohn's disease	20883	5956	14927	26192919 / Liu et al. 2015	European	https://www.ibdgenetics.org/	relatedness testing and PCs
	multiple sclerosis	115803	47429	68374	31604244 / International Multiple Sclerosis Genetics Consortium et al. 2019	European	https://www.ncbi.nlm.nih.gov/projects/gap/cgi-bin/study.cgi	first 5 PCs
	psoriasis	420461	4815	415646	30305743 / Bycroft et al., 2018	European	https://www.ncbi.nlm.nih.gov/gap/advanced_search/2TERM	25 PCs, self-reported sex, and genotyping array
	rheumatoid arthritis	58284	14361	43923	24390342 / Okada et al. 2013	European	http://plaza.umin.ac.jp/~yokada/datasource/software.htm	top 5 or 10 PCs
	ulcerative colitis	27432	6968	20464	26192919 / Liu et al. 2015	European	https://www.ibdgenetics.org/	relatedness testing and PCs

We appreciate the reviewer's suggestions for visual improvement of our work. We have accordingly plotted the sensitivity results for the TWAS-MR (Figure S5) and the drug target MR (Figure S6), as below:

Figure S5.

Figure S6.

In the MR analysis between cytokines, it is stated that “we used cis-acting variants as genetic instruments for our MR analyses.” However, it’s unclear how the cis variants overlap between cytokines/proteins. Do you mean that all cis variants of all cytokines were used?

We apologize for not being sufficiently clear in our manuscript. We used for each cytokine *only* its respective cis-instrument consisting of variants influencing the levels of this cytokine in the locus where its encoding gene is located. As these variants are coming from different genomic region per cytokine, there is no overlap in the variants used as cis-instruments between any of the 40 cytokines.

For the cross-cytokine analysis we calculated 40 MR analyses, for each we used all cis-acting variants as instruments and the clumped GWAS results of one cytokine each as outcome. We now rephrased the description in our methods as follows (Page 23, Line 1100f.): “For the (i) cross-cytokine MR analyses, for each cytokine, we used as genetic instruments cis-variants influencing its levels and located within the locus of their encoding gene.”

In the main text, it states, “Using cis-acting variants for G-CSF/CSF-3, CXCL9/MIG, and TNF-b from the UKBB proteomics dataset as instruments, we confirmed significant associations with asthma, Crohn’s disease, and multiple sclerosis, respectively.” Are these the only three proteins from UKBB that overlap with the 40 included in this study, and did they replicate all the associations identified in the MR with UKBB data? This context is crucial for interpreting the results.

We used the UKBB proteomics dataset to validate *only* the results of the cytokine-disease MR pairs that had the highest evidence of association: significant results in cis-MR analysis AND evidence of significant colocalization between cytokine levels and disease risk in the locus of the gene encoding the respective cytokine. These pairs included G-CSF/CSF-3 – asthma, CXCL9/MIG – G-CSF/CSF-3 – Crohn’s disease and TNF-b – multiple sclerosis. We changed the methods section clarifying this aspect (Page 23, Line 1108f.): “For the (ii) disease endpoint MR we validated the significant MR results that also showed significant colocalization between cytokine levels and disease risk in the locus of the cytokine gene by using genetic instruments derived from an independent UKBB proteomics dataset, selected based on the same criteria.”

3. There are also other potential errors, such as in the phrase “minimum allele frequency,” which I believe should be “minor allele frequency.”

We thank the reviewer for the careful read of our manuscript. We have now corrected the error replacing “minimum allele frequency” with “minor allele frequency” throughout the manuscript and the figures. We have also screened the entire manuscript to ensure lack of typos.

4. Finally, Supplementary Table S8 only provides data sources for the 40 cytokines but does not include those for the diseases used in the MR analysis.

Supplementary Table S8 provides the data sources for the disease datasets in the 2nd tab “S8 diseases” in the column “source download link” (see the screenshot in the answer to the 2nd comment).

Reviewers' comments:

Reviewer #2 (Remarks to the Author):

The authors have addressed my questions. I don't have anything further to add.